# The essential calcium channel of sperm CatSper is temperature-gated

Dilip K. Swain[1,2], Citlalli Vergara [1,2,4], Júlia Castro-Arnau [1,2,4] &
Polina V. Lishko [1,2,3] ✉

The flagellar calcium channel CatSper is essential for male fertility, as it regulates calcium influx to trigger the hyperactive motility required for sperm to fertilize the egg. Precise activation of CatSper is critical, as premature activation can impair sperm function. While optimal temperature is known to influence fertilization, its effect on CatSper remains unknown. By directly recording from mouse spermatozoa, we reveal that CatSper functions as a temperature-gated ion channel, with a thermal threshold of 33.5 °C and a temperature coefficient $Q_{10}$ of 5.1. Additionally, we show that physiological levels of spermine reversibly inhibit CatSper's temperature gating, protecting against premature activation. Our findings highlight for the first time the presence of the temperature-gating modality of CatSper and reveal the protective role of spermine, a major component of seminal plasma. These results emphasize the need to maintain testes below 34 °C for optimal fertility and advance understanding of CatSper regulation in male fertility.

Successful fertilization is crucial for species survival, and evolution has finely tuned this process by optimizing its timing, location, and environmental conditions. In species that fertilize internally, sperm must reach the egg at the right time after undergoing their final maturation, known as capacitation[1,2], in the oviduct. Capacitation enables sperm to develop hyperactive motility or hyperactivation – an essential motility pattern characterized by an asymmetrical flagellar bending that allows sperm to navigate through the oviduct to reach the egg[3–5]. Sperm that fail to undergo hyperactivation exhibit impaired fertilization ability. Heat has been shown to stimulate hyperactivation[4], a process driven by a flagellar calcium influx through the sperm-specific calcium channel, CatSper[6–10]. This ion channel is critical for male fertility, as CatSper-deficient rodents and human males are sterile[6–10]. CatSper activation is triggered by intracellular alkalinization[6,11], membrane depolarization, and, in primates, by exposure to progesterone[10–14]. However, despite extensive studies on CatSper activation, the role of temperature in its regulation remains largely unexplored.

While CatSper-mediated hyperactivation is essential for sperm fertility, premature CatSper activation must be avoided as it triggers ion channel degradation and renders sperm dysfunctional[15,16]. To prevent this, a safeguarding mechanism has likely evolved to keep CatSper closed until the sperm reach the egg. A similar mechanism must also prevent CatSper activation within the male reproductive tract, as its failure could lead to male infertility.

Here, we provide a comprehensive analysis of CatSper activation and deactivation, focusing on an underexplored factor influencing CatSper function- the physiological core body temperature. Despite mounting evidence that sperm hyperactivation is heat-dependent[4], and critical for fertilization[17], the role of temperature in CatSper regulation remains unexamined. Given the paramount importance of elevated temperatures (38–40 °C) in fertilization[17], we explored how physiological conditions mimicking the male and female reproductive tracts influence CatSper behavior.

Using direct recordings from wild-type and CatSper1[-/-] murine spermatozoa, we, for the first time, show that mammalian CatSper is a temperature-gated channel with a threshold activation at 33.5 °C. In addition, we examined sperm adaptation to temperature changes during capacitation and the protective role of spermine, an essential component of seminal plasma. We found that physiological levels of

[1]Department of Cell Biology and Physiology, WashU Medicine, Washington University in St. Louis, School of Medicine, St. Louis, MO, USA. [2]Center for the Investigation of Membrane Excitability Diseases (CIMED), WashU Medicine, St. Louis, MO, USA. [3]BJC Investigator Program, WashU Medicine, St. Louis, MO, USA. [4]These authors contributed equally: Citlalli Vergara, Júlia Castro-Arnau. ✉e-mail: lishko@wustl.edu

spermine matching those present in the seminal plasma inhibit Cat-Sper temperature-gating. In addition, we found that capacitated sperm adapt to the oviduct's warm environment, exhibiting reduced temperature sensitivity. Our findings reveal a remarkable mechanism by which CatSper fine-tunes its voltage-dependent, pH-sensitive, and temperature-gating properties, reshaping our understanding of temperature sensation in sperm. These results suggest that testicular externalization may have evolved, in part, to prevent premature Cat-Sper activation and ensure male fertility.

## Results

### Epididymal CatSper is a temperature-gated channel

To study the temperature regulation of CatSper, we applied the sperm patch-clamp technique to the murine spermatozoa isolated from cauda epididymis as described before[18,19]. Specifically, the seals were achieved in High Saline (HS) solutions followed by perfusion with divalent-free (DVF) cesium (Cs$^+$) - based solutions as previously described[11,18–20]. To ensure the currents are free of any potential contamination of potassium components, we also used a Cs$^+$-based pipette solution, which was shown to block sperm potassium currents[11,18,21]. Under DVF conditions, CatSper can pass Cs$^+$ faster than divalent ions and creates ~10-fold larger currents ($I_{Cs+}$), which allows for more accurate measurements[11,12,18–20].

After establishing the whole-cell mode of contact, the elevated temperatures were applied via a built-in in-line heater, ensuring a heat ramp from 22 °C to up to 43 °C. The mini probe detector was positioned near the recording sperm cell to ensure continuous recording of the exact temperature to which the cell was exposed. Additionally, to ensure the observed effect is CatSper-specific, we used CatSper1$^{-/-}$ mice to record from spermatozoa lacking any functional CatSper channels[6] (Fig. 1a–c). CatSper monovalent currents ($I_{Cs+}$) were recorded upon stimulation with voltage steps (Fig. 1a, b) to determine the conductance-membrane potential relationship (G-V curve) and current (I)-voltage (V) relationship (I-V curves). The voltage-ramp protocol (Fig. 1c) was also used to determine the time course of CatSper's heat response and pharmacology. When recordings were conducted at 22–24 °C with sperm internal pH of 7.4 (Fig. 1a (upper panel); Fig. 1c, d, black traces), $I_{Cs+}$ showed typical outward rectification, i.e., currents smaller at the negative membrane potentials than positive voltages. An average current density obtained at −80 mV was −61.2 ± 6.3 pA/pF (Fig. 1d), elicited by step recordings (Fig. 1a, top panel) and measured at the peak (yellow triangle). At the same temperature conditions, an average current density at + 80 mV was 180.0 ± 9.5 pA/pF. Slightly larger current densities of −143.5 ± 11.9 pA/pF at −80 mV (Fig. 1e) were obtained from ramp recordings (Fig. 1c; left panel, black trace). No significant $I_{Cs+}$ was observed when recorded from CatSper1$^{-/-}$ sperm (Fig. 1b, 1c (right panel, black); Supplementary Fig. S1a, b). The residual non-CatSper conductance observed did not exceed the current density of −6.1 ± 0.5 pA/pF at −80 mV at 24 °C (Supplementary Fig. S1b).

However, when CatSper1$^{+/+}$ sperm were subjected to gradually increasing temperatures up to 40 °C, $I_{Cs+}$ significantly increased, particularly in the negative, i.e., inward direction (Fig. 1a (lower panel), 1c (left panel; red), and 1 d, red). A significant temperature-stimulated increase was detected with voltage step protocols (Fig. 1a), which is also reflected in the corresponding I-V curves obtained at the peak (yellow triangles; Fig. 1d, red). An average current density now reached −220.3 ± 41.3 pA/pF at −80 mV (Fig. 1d, red) and was obtained as above, from step-recordings (Fig. 1a) and measured at the peak (yellow triangle). At the same heat conditions, even larger current densities of −281.8 ± 20.0 pA/pF at −80 mV (Fig. 1e) were obtained from ramp recordings (Fig. 1c; left panel, red). This heat activation of $I_{Cs+}$ was completely reversible (Fig. 1e, f). No heat-activated conductance was detected at 38 °C from any of the thirteen assessed CatSper1$^{-/-}$ sperm (Fig. 1b (lower panel); 1c (right panel, red); Supplementary Fig. S1a–c).

The remaining non-CatSper current density observed at 38 °C from CatSper1$^{-/-}$ sperm was not different from the one observed at 24 °C and was −7.1 ± 0.5 pA/pF at −80 mV (Supplementary Fig. S1b). CatSper1$^{-/-}$ sperm were viable, which is evident from various residual non-CatSper currents that appeared upon sperm exposure to indicated extracellular monovalent ions (Supplementary Fig. S1d).

Moreover, when CatSper was partially inhibited in wild-type sperm with NNC 55- 0396 (Supplementary Fig. S1e), no heat-activated potentiation was observed. Even when the temperature ramps were exceeding 40 °C (Supplementary Fig. S1f, g; blue dotted line and the arrow), no additional conductance was detected.

The mechanism of CatSper heat-activation seems to be related to its voltage dependence as observed by a dramatic leftward shift of its G-V curve from the midpoint ($V_{1/2}$) of −9.4 mV at 22 °C (Fig. 1g, black) to $V_{1/2} = −85.5$ mV at 38 °C (Fig. 1g, red). G-V curve reflects a relative percentage ("1" on the y-axis corresponds to 100%) of the channels in the open state to the corresponding membrane voltages, shown on the x-axis. It can be measured using amplitudes of the tail currents, as shown in Fig. 1a (acquired at points indicated by green triangles). The typical membrane voltages of healthy non-capacitated murine spermatozoa are within −42 to −60 mV[22–24], as shown by the shaded blue area in Fig. 1g. Therefore, exposure to 38 °C stabilizes the open state of the CatSper.

The same behavior of the CatSper channel was observed with divalent $I_{Ba}^{2+}$ where Ba$^{2+}$ ions were used to study divalent ion influx to avoid Ca$^{2+}$-dependent inactivation of CatSper (Fig. 2a–e). $I_{Ba}^{2+}$ were stimulated by either voltage steps (Fig. 2a), or by voltage ramps (Fig. 2b). A similar heat ramp was applied from 22 °C to 40 °C followed by a reversible cooling to 35 °C (Fig. 2c). The average current densities for $I_{Ba}^{2+}$ acquired at the peak at −80 mV (Fig. 2a, yellow triangle) were −4.3 ± 0.2 pA/pF at 24 °C, and −22.8 ± 7.2 pA/pF at 38 °C (Fig. 2d), and data were used to build the I-V curve (Fig. 2d). G-V curves for $I_{Ba}^{2+}$ (Fig. 2e) were acquired similarly as mentioned above for $I_{Cs}^+$ by recording amplitudes of the tail currents (Fig. 2a, green triangles). Midpoint activation for divalent currents showed a similar profound shift from $V_{1/2} = + 45.3$ mV (Fig. 2e, black) to −20.8 mV at 38 °C (Fig. 2e, red).

To determine the temperature sensitivity of CatSper, the 10-degree temperature coefficient $Q_{10}$ was measured using $I_{Ba}^{2+}$ elicited by the voltage ramps (Fig. 2b). The current amplitudes were sampled at a fixed timepoint of 400 ms that roughly corresponds to −20 mV (Fig. 2b; gray triangle). The $Q_{10}$ of an ion channel represents the fold increase in the current with a 10 °C rise in temperature. For CatSper, it can be determined by calculating the inverse slope of a log (normalized $I_{Ba}^{2+}$) plotted against 1/T, where T is the absolute temperature in K$^{-1}$ (Fig. 2f). $Q_{10}$ of the channels that are not temperature-gated is typically around 1.5. However, murine sperm subjected to a gradual heating protocol initially displayed a slow activation phase with a $Q_{10}$ of 2.7 ± 0.1 ($n = 7$ cells), followed by a steeper phase with a $Q_{10}$ of 5.01 ± 0.27 ($n = 7$ cells) (Fig. 2f and Supplementary Fig. S2). The transition point between these phases is known as the thermal threshold ($T_h$) and was observed at 33.5 °C, which, interestingly, is the temperature of the murine scrotum.

This data suggests murine CatSper possesses the property of a temperature-gated ion channel, at least when recorded from non-capacitated, freshly isolated epididymal spermatozoa subjected to conditions mimicking that of the testicular environment. This ability would trigger premature activation of CatSper upon testicular heating, so, logically, heat exposure should be minimized.

### Heat-activation of CatSper is attenuated by acidic conditions

After sperm are transferred to the epididymis, they are stored within the epididymal duct under acidic conditions, with a pH below 6.0. External acidification usually precedes internal acidification, as sperm adjust their pH to the external environment[25]. Therefore, we explored

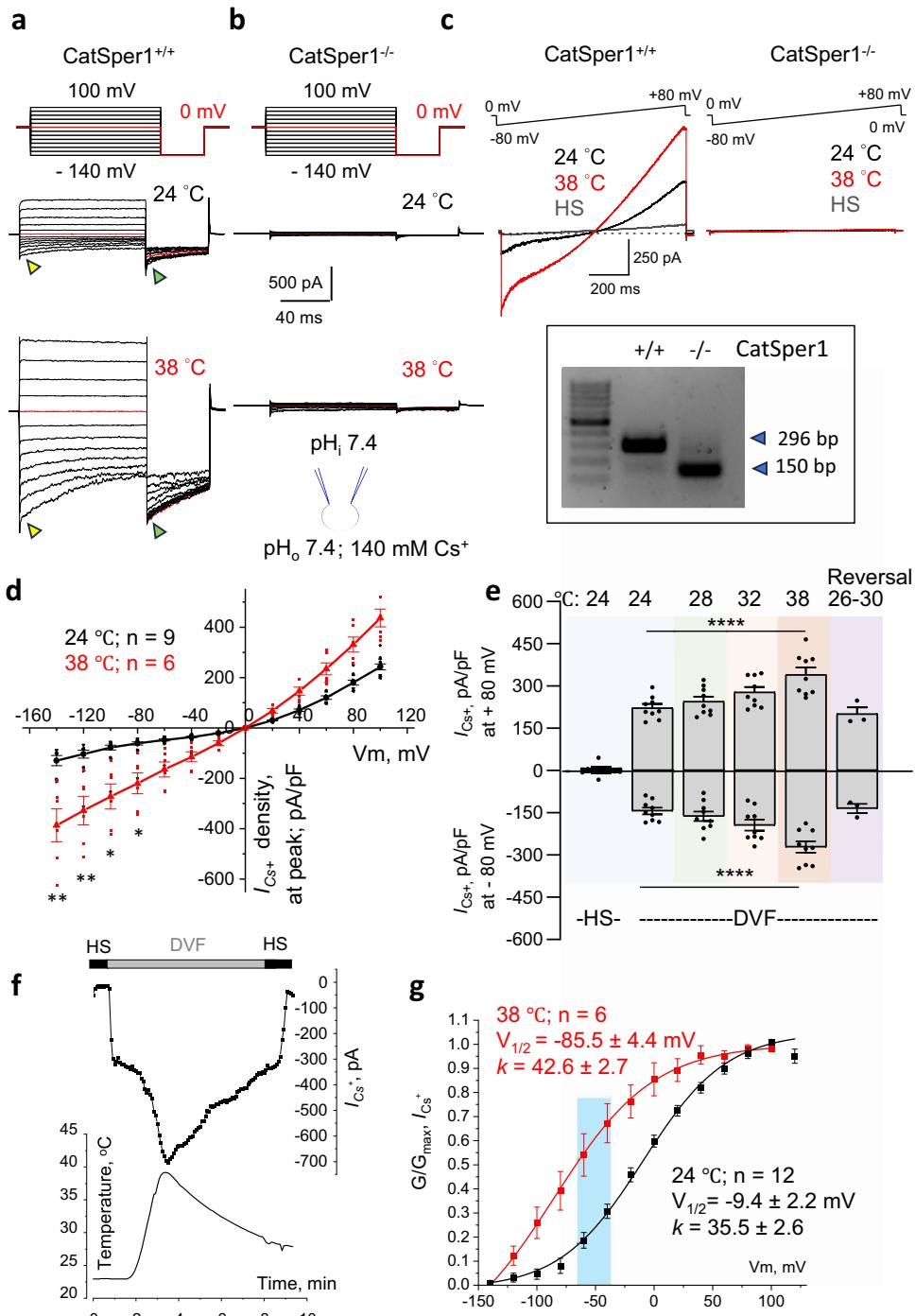

**Fig. 1 | Murine epididymal CatSper is activated by heat. a** Representative whole-cell patch clamp recordings from CatSper1$^{+/+}$ murine sperm in response to indicated voltage steps at 24 °C (upper panel) and 38 °C (lower panel). Red traces indicate currents recorded at 0 mV. Triangles point to corresponding peak $I_{Cs}^+$ amplitudes (yellow) and tail $I_{Cs}^+$ amplitudes (green). **b** Representative recordings from CatSper1$^{-/-}$ sperm lacking $I_{Cs}^+$ at indicated temperatures in response to similar voltage steps. **c** $I_{Cs}^+$ in response to voltage ramps at 24 °C (black) and 38 °C (red), recorded from CatSper1$^{+/+}$ (left panel) and CatSper1$^{-/-}$ (right panel) murine sperm. Insert shows mouse genotyping. Currents in the HS solution show baseline conductance. **d** Current-voltage relationships (I-V curves) were calculated from amplitudes shown on (**a**) as indicated by yellow triangles. Exposure to heat led to a significant increase in $I_{Cs}^+$. P-values were as follows: 0.002 (at −140 mV), 0.0017 (at −120 mV); 0.01 (at −100 mV and −80 mV). **e** Murine $I_{Cs}^+$ densities (mean values ± S.E.M.; pA/pF) from CatSper1$^{+/+}$ sperm obtained at −80 mV and +80 mV at 24 °C, 28 °C, 32 °C, and 38 °C ($n = 9$), as well as the same cells' $I_{Cs}^+$ densities obtained after

cooling (reversal). $I_{Cs}^+$ densities obtained at −80 mV were as follows: −143.54 ± 11.86 (at 24 °C); −162.20 ± 17.17 (at 28 °C); −194.44 ± 19.54 (at 32 °C); and −271.81 ± 19.97 (at 38 °C). Reversal $I_{Cs}^+$ density at −80 mV was −134.79 ± 16.32; $n = 3$. $I_{Cs}^+$ densities obtained at +80 mV were as follows: 222.93 ± 13.16 (at 24 °C); 245.84 ± 16.31 (at 28 °C); 278.37 ± 17.66 (at 32 °C); and 340.85 ± 24.48 (at 32 °C). P-values for both outward and inward current densities were < 0.0001. **f** Time-course of $I_{Cs}^+$ response (upper panel) to heat ramp (lower panel) recorded from CatSper1$^{+/+}$ freshly isolated murine sperm was stable and reversible. **g** Conductance-membrane potential relationships (G-V curve) calculated from tail current amplitudes (green triangles; (**a**)) at 24 °C (black) and 38 °C (red) show midpoint activation shift (V$_{1/2}$) from −9.4 mV to −85.5 mV; $k$ indicates slope factor. The shaded blue area indicates the physiological range of membrane voltages for non-capacitated murine spermatozoa. Data are mean values ± S.E.M., n corresponds to the number of cells used. Insert shows the main conducting ion and pH of the solutions used for this figure.

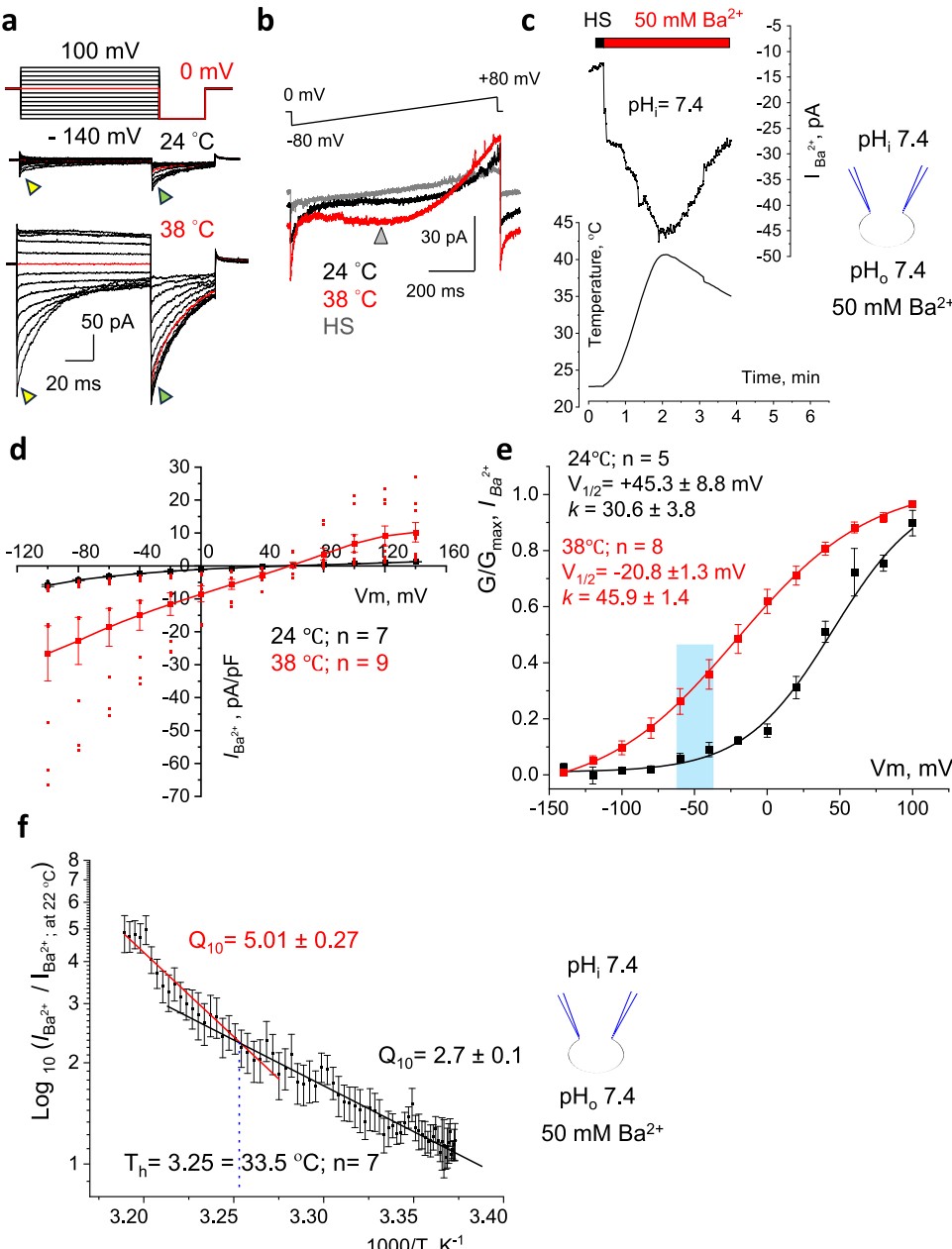

**Fig. 2 | Murine epididymal CatSper is a temperature-gated ion channel.**
**a** Representative $I_{Ba^{2+}}$ from CatSper1[+/+] sperm in response to indicated voltage steps. Exposure to 38 °C led to a significant increase in $I_{Ba^{2+}}$. Triangles point to corresponding peak $I_{Ba^{2+}}$ amplitudes (yellow) and tail $I_{Ba^{2+}}$ amplitudes (green).
**b** Representative $I_{Ba^{2+}}$ stimulated by a voltage ramp at 24 °C (black) and at 38 °C (red). A gray triangle indicates a timepoint corresponding to - −20 mV. **c** The heat response of $I_{Ba^{2+}}$ recorded from wild-type sperm was stable and reversible. **d** I−V curves were calculated from peak amplitudes shown on (**a**), as pointed by yellow triangles. **e** G-V curves for $I_{Ba^{2+}}$ were calculated from tail amplitudes shown on (**a**) as indicated by green triangles. The shaded blue area indicates the physiological range of membrane voltages for non-capacitated spermatozoa. **f** The current-temperature relationship was determined using $I_{Ba^{2+}}$ amplitudes sampled at − 20 mV (gray triangle) and stimulated by temperature ramps from 24 °C to 41 °C. The linear fits of two phases of thermal response were used to determine $Q_{10}$ and a thermal threshold ($T_h$). Data are mean values ± S.E.M, n corresponds to the number of cells used. Inserts show the main conducting ions and pH of the solutions used.

whether the heat-activation of CatSper is preserved under intracellular acidification (Fig. 3), where intracellular pH (pH$_i$) was kept at 6.0. Given known CatSper pH-sensitivity and its inhibition by acidic pH[9,11,26], initial CatSper currents recorded at 22–24 °C using voltage steps were expectedly smaller (Fig. 3a, upper panel; Fig. 3b, black and Supplementary Fig. S3a–f) in comparison to the currents recorded at pH$_i$ 7.4 (Fig. 1a, upper panel; Supplementary Fig. S3a, c–f). An average $I_{Cs+}$ density at −80 mV and pH$_i$ = 6.0, at 22–24 °C was only −48.4 ± 6.4 pA/pF recorded at the peak (Fig. 3a; yellow triangles and Fig. 3b). However, heat exposure still produced CatSper activation (Fig. 3a (lower panel); 3b (red)). Given initial smaller $I_{Cs+}$ obtained at

acidic conditions, inactivating $I_{Cs+}$ reached baseline faster than with normal pH (Supplementary Fig. S3a–f). Thus, no significant increase in $I_{Cs+}$ was observed upon heat exposure when measured at a steady state (Fig. 3c; Fig. 3a, blue triangles; Supplementary Fig. S3g, h). No significant changes in the shape of the $I_{Cs+}$ inactivation were observed when tail currents recorded under acidic or normal conditions were compared (Supplementary Fig. S3a–f).

G-V curve showed a small nonsignificant leftward shift upon heat exposure from midpoint activation of + 43.2 mV to + 24.6 mV (Fig. 3d), which is insufficient to ensure CatSper's open state. Temperature-activated CatSper currents were also completely reversible (Fig. 3e).

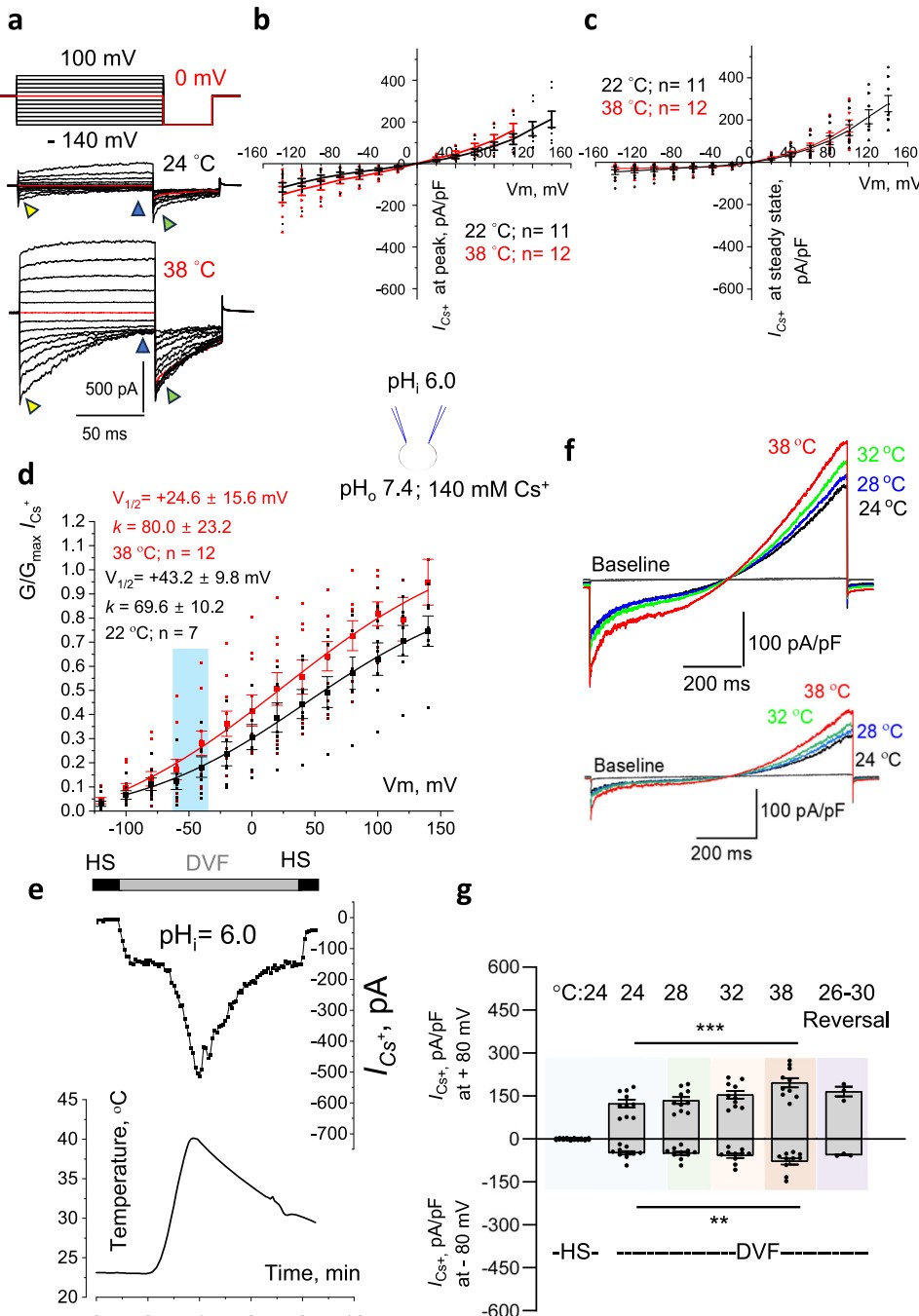

**Fig. 3 | Intracellular acidic pH protects murine CatSper from heat activation.** **a** Representative $I_{Cs^+}$ recording in response to voltage steps at 24 °C (upper panel) to 38 °C (lower panel). Exposure to 38 °C led to a noticeable but fast-inactivating increase in the inward currents. **b**, **c** I–V curves were calculated from amplitudes shown on (a) at the peak (yellow triangles) or the steady state (blue triangles). **d** G-V curves were calculated from (**a**) tail current amplitudes (green triangles) at 22–24 °C and 38 °C. A non-significant midpoint activation shift ($V_{1/2}$) from +43.2 mV to +24.6 mV was insufficient for CatSper activation. Slope factor $k$ indicates shallow voltage dependence. **e** Representative heat response of $I_{Cs^+}$ recorded from wild-type sperm kept at intracellular pH 6.0 is brief, inactivating, and reversible. **f** Two representative $I_{Cs^+}$ recordings in response to indicated temperatures demonstrating variability in heat response. The baseline indicates $I_{Cs^+}$ in HS solution. **g** $I_{Cs^+}$ densities (mean values ± S.E.M; pA/pF) from murine sperm stimulated by a voltage ramp and sampled at −80 mV and +80 mV, at 24 °C, 28 °C, 32 °C, and 38 °C ($n = 10$). $I_{Cs^+}$ densities obtained after the same cells were cooled down (reversal; $n = 3$) are also shown. Specifically, at −80 mV $I_{Cs^+}$ densities were: −48.42 ± 6.39 (at 24 °C); −50.01 ± 6.37 (at 28 °C); −58.48 ± 7.58 (at 32 °C); and −78.56 ± 11.05 (at 38 °C). At +80 mV $I_{Cs^+}$ densities were: 123.89 ± 13.23 (at 24 °C); 134.47 ± 12.17 (at 28 °C); 154.24 ± 13.55 (at 32 °C); and 196.79 ± 16.17 (at 38 °C). Reversal $I_{Cs^+}$ density was −55.54 ± 2.12 pA/pF (at −80 mV). At −80 mV $p = 0.0022$ and at +80 mV, $p = 0.0002$. Insert shows the main conducting ion and pH of the solutions used.

When $I_{Cs^+}$ were recorded using voltage ramps, significant variability in heat response was observed among the sperm population (Fig. 3f, g). However, these currents were consistently smaller compared to those recorded under normal intracellular pH conditions (Figs. 1c, 3f), and averaged $I_{Cs^+}$ densities measured at a steady state (blue triangles)

at both 22–24 °C and 38 °C with pH$_i$ = 6.0 were significantly lower than $I_{Cs^+}$ densities measured at 38 °C with pH$_i$ = 7.4 (Supplementary Fig. S3g, h). Thus, while elevated temperature can induce brief CatSper activation in the epididymal environment, the acidic pH provides robust protection against prolonged CatSper activation.

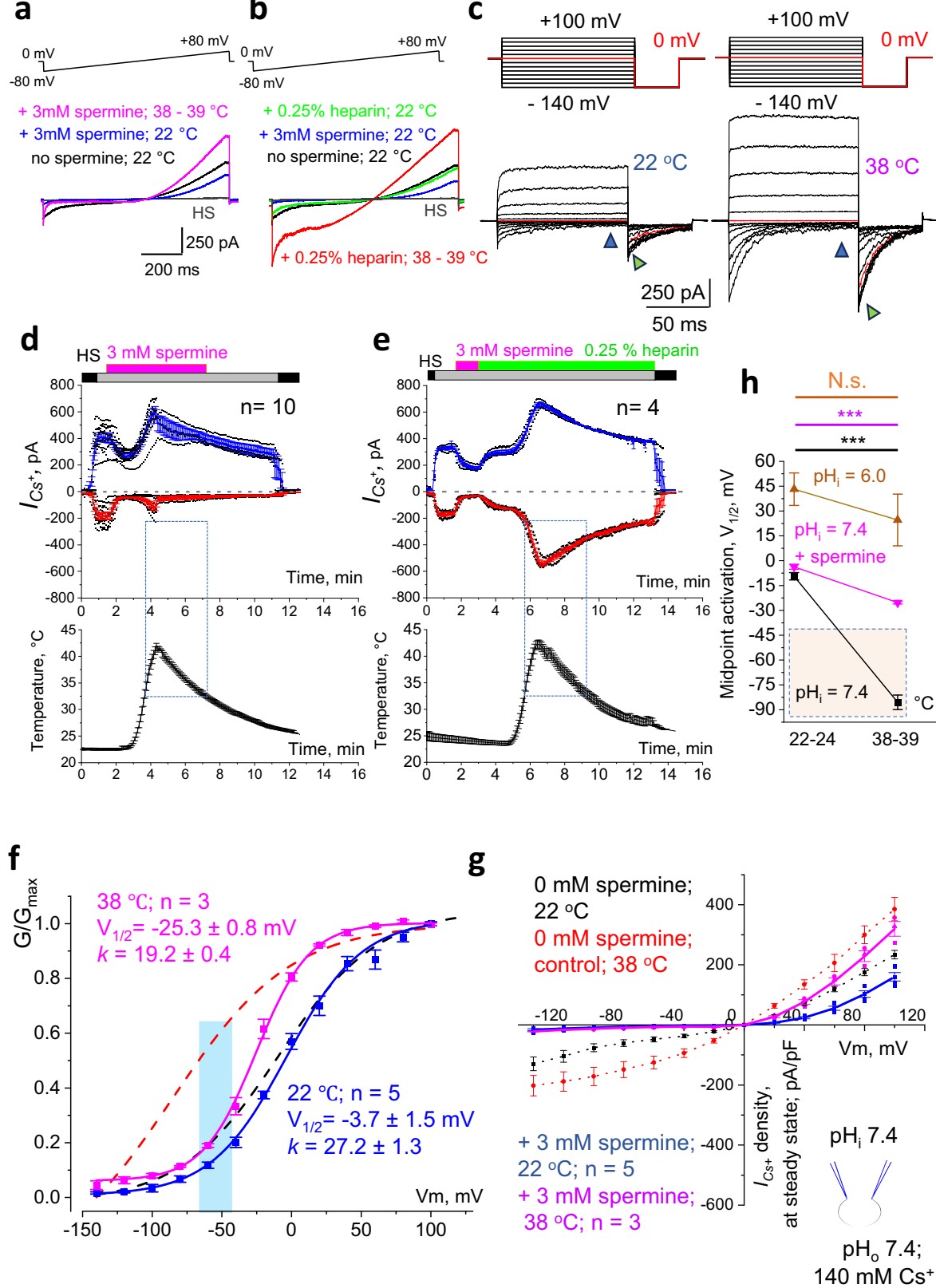

## Spermine inhibits CatSper and prevents its temperature activation by changing CatSper's voltage-dependence

Upon deposition into the female reproductive tract, spermatozoa are immediately exposed to 37 °C, and their intracellular pH (pH$_i$) rises to 7.4, creating optimal conditions for instant CatSper activation. Therefore, an additional protective mechanism must exist to prevent

premature CatSper activation. After ejaculation, spermatozoa are mixed with seminal plasma, which contains a cocktail of components known to play a crucial role in preserving sperm functionality[27]. However, their interactions with sperm ion channels were minimally known. Seminal plasma is particularly enriched in spermine, which reaches 3 mM concentrations[28,29]. Spermine is known to affect the

**Fig. 4 | Spermine inhibits CatSper and protects it from temperature activation.** **a** Representative $I_{Cs}^+$ recorded from wild-type spermatozoa in the presence of spermine in response to 22 °C (blue) and 38 °C (magenta). **b** Spermine inhibition is reversed with 0.25% heparin at 22 °C (green) and 38 °C (red). **c** Representative $I_{Cs}^+$ in response to indicated voltage steps in the presence of spermine. Note the fast inactivation of the responses at 38 °C (right panel). Triangles point to corresponding $I_{Cs}^+$ amplitudes at the steady state (blue) and tail $I_{Cs}^+$ amplitudes (green). **d** Time-course of $I_{Cs}^+$ reversible response (upper panel) to heat ramp (lower panel) recorded at + 80 mV (blue) and −80 mV (red) in the presence of 3 mM spermine. **e** Time-course of $I_{Cs}^+$ reversible response after spermine was removed with 0.25 % heparin and stimulated by temperature ramp as in (**c**). Dotted areas show maximal CatSper activation upon heat exposure above 34 °C. The presence of spermine prevents inward $I_{Cs}^+$ from reaching the same amplitude (**d, e**). Data are presented as mean values ± S.E.M., n corresponds to the number of cells used for (**d, e**). **f** G-V curves were calculated from $I_{Cs}^+$ tail currents (**c**, green triangles) in the presence of spermine at 22 °C (blue) and 38 °C (magenta). For comparison, G-V curves in the absence of spermine at 22 °C (black) and 38 °C (red) are shown as dashed lines representing G-V curves from Fig. 1g. **g** I-V curves were calculated at the steady state in the presence of spermine (**c**, green triangles) at 22 °C (blue) and 38 °C (magenta). Dotted I-V curves show recordings in the absence of spermine, as in Supplementary Fig. S3g. **h** Spermine exposure (purple) but not acidic pH (brown) significantly shifted $I_{Cs}^+$ midpoint activation upon heat. However, the spermine-induced shift resulted in $V_{1/2}$ change from − 3.7 mV to only − 25.3 mV, which is still outside of the physiological range of membrane voltages for non-capacitated spermatozoa, as shown by the dotted and shaded area. The shift of $V_{1/2}$ upon heat for nonexposed control spermatozoa recorded with normal (pH = 7.4) pH (black) is shown for comparison. Data are presented as mean values ± S.E.M., n corresponds to the number of cells used. Data are cumulative from Figs. 1g, 3d, and 4f. P-values were as follows: 0.001 (spermine; magenta) and 0.0013 (control; black).

behavior of various ion channels[30,31], however, its effect on CatSper was also unknown. To explore the effect of spermine on CatSper behavior, we exposed epididymal murine spermatozoa to 3 mM spermine, introduced to the cells via extracellular application using continuous perfusion (Fig. 4a–e).

Within a few seconds, spermine inhibited $I_{Cs+}$ (Fig. 4a) and altered its temperature sensitivity. Heat activation produced a small and sharp increase in the inward current, followed by almost immediate inactivation of the inward $I_{Cs+}$ (Fig. 4c, right panel). This inhibition did not wash out even after 4 minutes of continuous perfusion with DVF solution (Fig. 4d, Supplementary Fig. S4a, b). Interestingly, outward $I_{Cs+}$ was less affected by spermine (Fig. 4a, c, d and Supplementary Fig. S4a, b). Only after a longer incubation in 0.25% heparin- the known factor for spermine removal involved in the capacitation process[32] - the temperature sensitivity was partially restored (Figs. 4b, 4e, and Supplementary Fig. S4c–e).

Spermine application produced the following effects: firstly, by inhibiting inward $I_{Cs+}$ (Fig. 4a, d), spermine exposure resulted in stronger $I_{Cs+}$ outward rectification at 38 °C (Fig. 4a). The latter profoundly decreased the CatSper slope factor (k), from 42.6 (Fig. 1g) to 19.2 (Fig. 4f). In addition, spermine shifted the midpoint activation upon heat exposure from −85.5 mV (spermine absence; Fig. 1g) to −25.3 mV (spermine presence; Fig. 4f), putting CatSper outside of its activation zone (Fig. 4f, S4f). Thus, by acting as a CatSper inhibitor, spermine alters CatSper's voltage- and temperature-gating and increases CatSper outward rectification (Fig. 4g). Taken together, these results suggest that both intracellular acidification and spermine exposure serve as physiological mechanisms to prevent CatSper premature activation. After leaving the acidic and cold epididymal environment and arriving in the hot female reproductive tract, spermatozoa benefit from spermine protection. Specifically, spermine efficiently shifts CatSper midpoint activation outside its physiological ranges of cell membrane potentials (Fig. 4h), keeping CatSper firmly shut.

## CatSper in spermatozoa subjected to capacitated conditions loses temperature gating

However, after prolonged exposure to the oviductal environment, sperm cells lose spermine's protection, as they gradually undergo capacitation[5,33,34]. Since spermine removal restored CatSper temperature-gating in vitro (Fig. 4b, e and Supplementary Fig. S4c–e), one would hypothesize that spermine removal during natural capacitation[32,35,36] may also affect CatSper in the same manner. To explore to what extent capacitated CatSper responds to heat, we recorded from murine spermatozoa subjected to 45 and > 90 min in vitro capacitation.

Specifically, epididymal murine spermatozoa were incubated in a capacitation solution at 37 °C and 5% $CO_2$, as described in Methods and in ref. 18. Timed capacitation was performed at specific intervals, followed by patch-clamp experiments. Capacitated spermatozoa were identified by their visible hyperactivated motility (Supplementary Movies 1–3 and Supplementary Fig. S5A–C). While kept at room temperature, $I_{Cs+}$ were slightly smaller in capacitated cohorts of spermatozoa, and CatSper currents gradually decreased as capacitation progressed up to 135 minutes (Fig. 5a (black); b (upper panel); c (black); Supplementary Fig. S5d, e). An average $I_{Cs+}$ density for monovalent currents at −80 mV was −98.2 ± 5.1 pA/pF (Fig. 5c and Supplementary Fig. S6a). $I_{Ba2+}$ were also visibly smaller for the capacitated cohort of murine sperm (Fig. 5d (black); e (upper panel); f (black)). However, heat exposure produced only a mild and reversible increase in capacitated $I_{Cs+}$ (Fig. 5a (red); b (lower panel); c (red); Supplementary Fig. S6a, b, S6c). Divalent conductance $I_{Ba2+}$ resembled the same trend (Fig. 5d (red); e (lower panel); f (red); Supplementary Fig. S6d). Interestingly, capacitated spermatozoa always demonstrated significantly higher membrane capacitance, suggestive of an increased membrane surface (Fig. 5g, h).

Midpoint activation for monovalent ions in capacitated spermatozoa retained a midpoint leftward shift from + 2.5 mV to −47 mV upon heat exposure (Fig. 5i). However, capacitated spermatozoa are known to hyperpolarize their membrane to −70 mV[23,24] (Fig. 5i, green shaded area) which would put capacitated CatSper further away from its activation range. Moreover, the midpoint activation shift for divalent ions was even less dramatic: with a tiny change from −9.1 to −9.6 mV (Fig. 5j and Supplementary Fig. S6e). Therefore, by attenuating CatSper temperature sensitivity, capacitated sperm cells adapt to warm temperatures in the oviduct. This adaptation makes CatSper gating in capacitated sperm temperature-independent, thus, allowing sperm to survive longer in the oviductal environment and await activation cues provided by the ovulated egg.

These results indicate that testicular spermatozoa kept at normal pH would be the most vulnerable to elevated temperatures. However, upon sperm transition into the epididymis, its acidity protects from heat-induced CatSper activation. Yet, sperm may become vulnerable again upon deposit into the female genital tract, where they are exposed to 38 °C and a variety of pH gradients, including slightly alkaline pH of the oviduct. To keep CatSper inactive in these conditions, an initial layer of protection is executed by spermine. However, upon capacitation, CatSper loses its temperature sensitivity and adapts to the oviductal environment, only to become active again, perhaps upon exposure to the factors provided by the ovulated egg or cumulus cells[37–40].

## CatSper is the main temperature-gated cation channel in murine spermatozoa

While CatSper is the major sperm calcium channel, another temperature-gated channel, TRPV4 was reported in sperm[41,42] and may participate in heat response. To explore mRNA expression levels for other temperature-regulated ion channels, we have obtained sperm transcriptome (Fig. 6a, b) that indicated the presence of TRPV4,

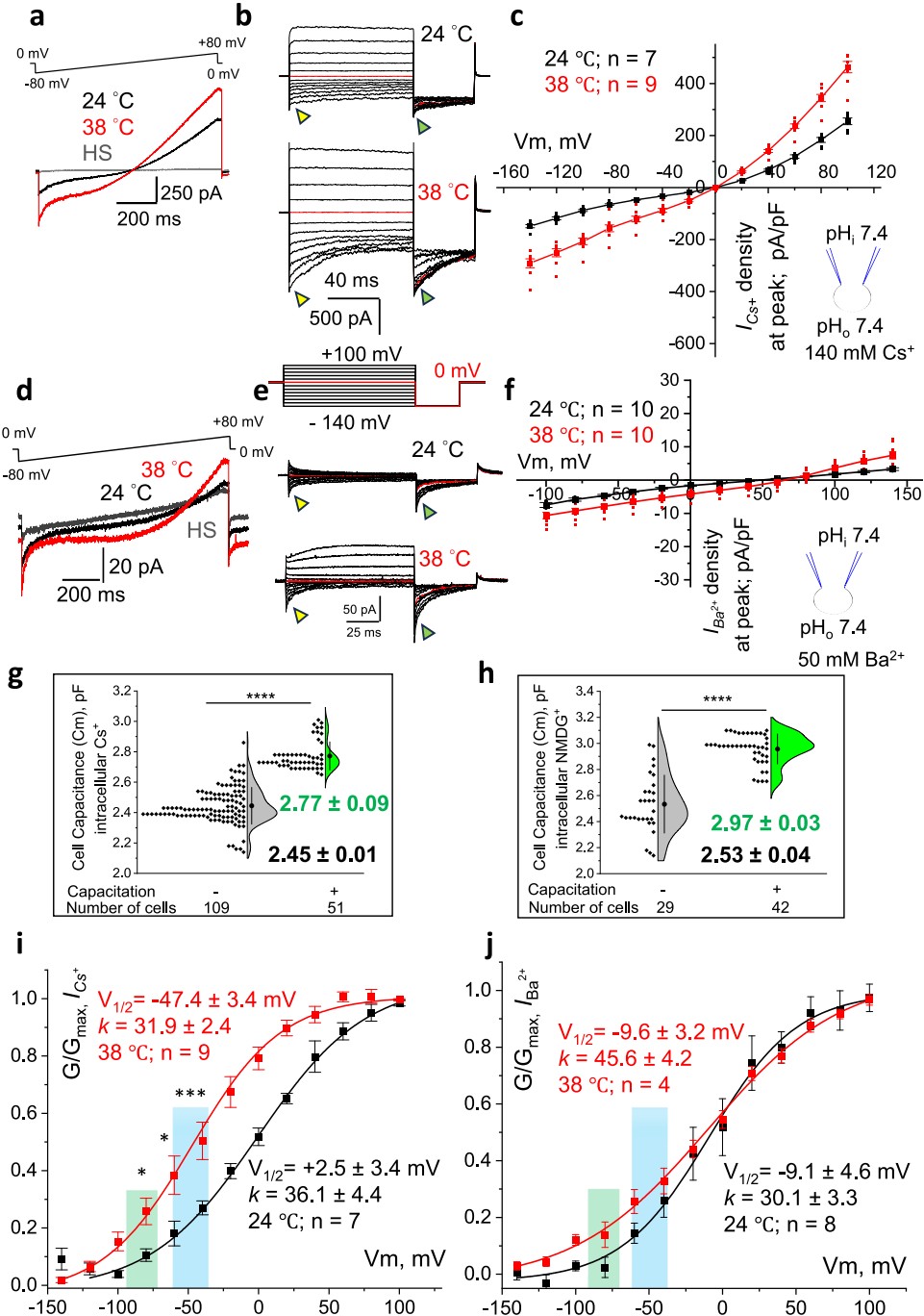

**Fig. 5 | Sperm capacitation reduces CatSper's response to heat. a** Representative $I_{Cs}^+$ from wild-type capacitated sperm at 24 °C (black) and 38 °C (red) in response to voltage ramp. The baseline indicates the HS solution. **b** Representative $I_{Cs}^+$ from capacitated sperm in response to indicated voltage steps. Exposure to heat increased $I_{Cs}^+$. **c** I–V curves calculated from amplitudes shown on (b, yellow triangles). Exposure to 38 °C led to a noticeable increase in $I_{Cs}^+$. Insert shows the main conducting ion and pH of the solutions used in (a–c). **d** Representative $I_{Ba}^{2+}$ from capacitated sperm at 24 °C (black) and 38 °C (red) in response to a voltage ramp. **e** Representative $I_{Ba}^{2+}$ from capacitated sperm in response to indicated voltage steps. **f** I–V curves were calculated from amplitudes shown on (**e**, yellow triangles). Unlike $I_{Cs}^+$, exposure to 38 °C led to only a minimal increase in $I_{Ba}^{2+}$, due to fast inactivation of the responses. Insert shows the main conducting ion and pH of the solutions used in (**d**–**f**). **g** Capacitated spermatozoa show larger capacitance

(Cm) for $I_{Cs}^+$: $Cm_{non-capacitated} = 2.45 \pm 0.01$ pF, $n = 109$, vs $Cm_{capacitated} = 2.77 \pm 0.09$ pF, n = 51. **h** Capacitated spermatozoa also showed increased Cm for $I_{Ba}^{2+}$: $Cm_{non-capacitated} = 2.53 \pm 0.04$ pF, n = 29, vs $Cm_{capacitated} = 2.97 \pm 0.03$ pF, $n = 42$. *P*-values were 0.0001 for both (**g** and **h**). **i** G-V curve calculated from tail amplitudes (b, green triangles) at 24 °C (black) and 38 °C (red) shows a reduced leftward shift. *P*-values were 0.015 (at −80 mV), 0.04 (at −60mV), and 0.001 (at −40mV). **j** G-V curves for $I_{Ba}^{2+}$ were calculated from tail amplitudes shown on (**e**, green triangles) at 24 °C (black) and 38 °C (red). Insert shown on (**c**) refers to the main conducting ion and pH of the solutions used in (**a**–**g**), and (i), while the insert shown on (**f**) refers to the same parameters used in (d)-(f), (h) and (j). Data are mean values ± S.E.M., n corresponds to the number of cells used. Sperm cells analyzed for this data set were subjected to 45 min of capacitation.

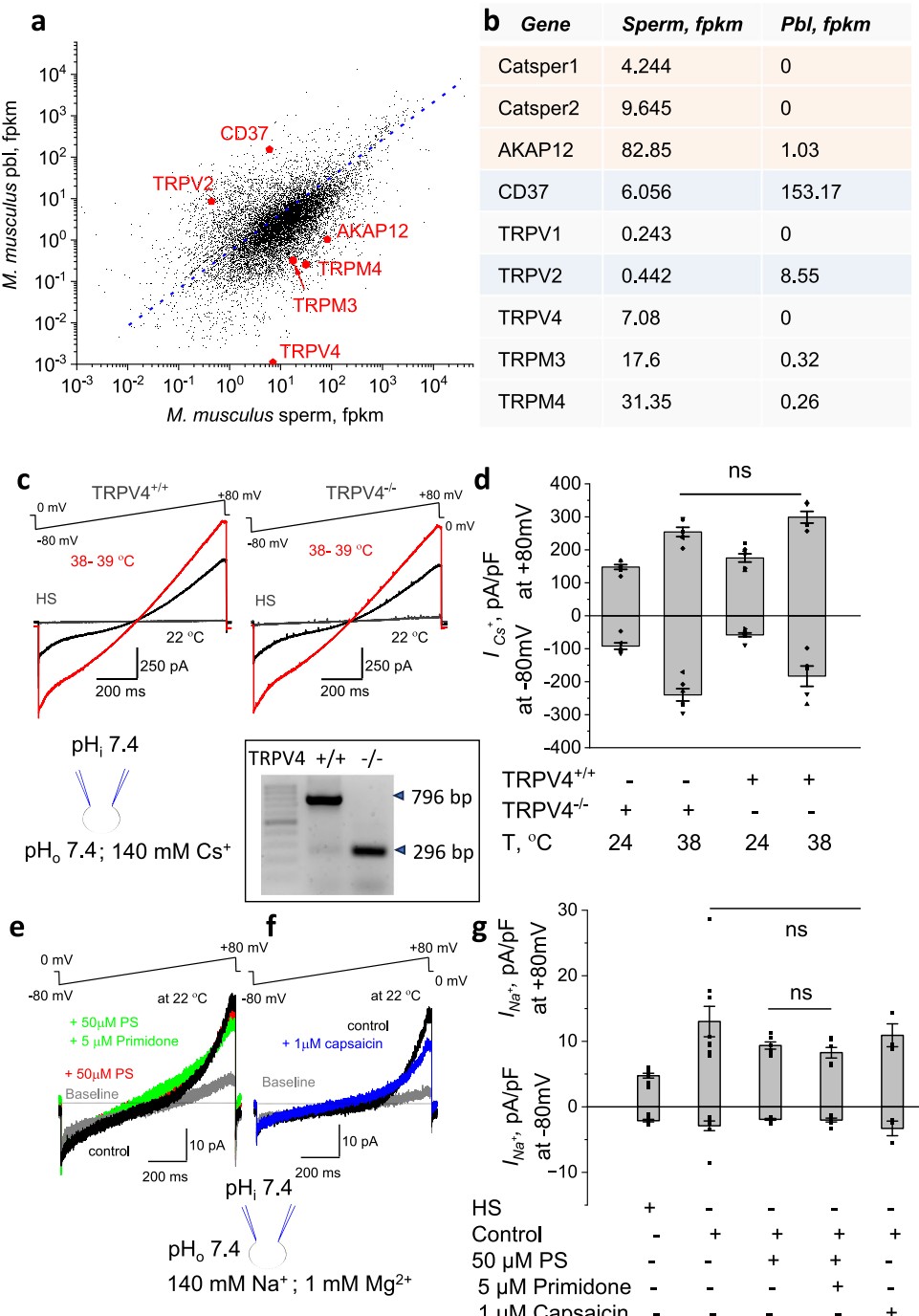

| Gene | Sperm, fpkm | Pbl, fpkm |
|---|---|---|
| Catsper1 | 4.244 | 0 |
| Catsper2 | 9.645 | 0 |
| AKAP12 | 82.85 | 1.03 |
| CD37 | 6.056 | 153.17 |
| TRPV1 | 0.243 | 0 |
| TRPV2 | 0.442 | 8.55 |
| TRPV4 | 7.08 | 0 |
| TRPM3 | 17.6 | 0.32 |
| TRPM4 | 31.35 | 0.26 |

TRPM4, and TRPM3 (NCBI; GEO accession number GSE290696). To explore their contribution to sperm heat response, we recorded from TRPV4$^{-/-}$ murine sperm (Fig. 6c, d and Supplementary Fig. S7a, b). No significant differences were observed between sperm from wild-type and TRPV4$^{-/-}$ mice (Fig. 6c, d and Supplementary Fig. S7a, b). Additionally, pharmacological interrogation of murine sperm cells with pregnenolone sulfate (PS), which acts as a TRPM3 agonist (Fig. 6e, g), or primidone (TRPM3 antagonist; Fig. 6e, g), or capsaicin (TRPV1 agonist; Fig. 6f, g), ruled out the presence of a functional TRPM3 or TRPV1 channels. To confirm PS and primidone activity, these compounds were successfully tested using recombinantly expressed human TRPM3 in HEK293 (Supplementary Fig. S8a, b).

TRPM4 was unlikely to contribute to the observed temperature-related effect on $I_{Cs+}$, as this ion channel requires intracellular calcium for activation. In our conditions, intracellular calcium was heavily buffered with BAPTA and/or EGTA resulting in less than 50 nM of free calcium, which is insufficient for TRPM4 activity. Also, TRPM4 preferentially conducts Cs$^+$ or Na$^+$, and neither of these currents was abundant in sperm lacking CatSper (Fig. 1b, c (right panel), Supplementary Fig. S1a, b, d, g). Only residual non-CatSper conductance was observed when CatSper was either absent (Supplementary Fig. S1a, S1b) or pharmacologically inhibited (Supplementary Fig. S1d), however, this remaining conductance was not temperature sensitive.

Overall, our results point to a precise-tuning mechanism of mammalian CatSper adaptation during the sperm journey from a colder testicular environment with alkaline pH to an acidic lumen of the epididymis, and eventually to a warmer and neutral-to-alkaline pH environment of the oviduct. Therefore, conserved fine-tuning between CatSper pH-sensitive, temperature-gating, and voltage-dependent properties, in the presence and absence of spermine, sperm cells

**Fig. 6 | Other temperature-gated channels in murine sperm do not influence CatSper temperature-gating. a** TRPV4, TRPM3, and TRPM4 are present in murine sperm transcriptome. mRNA sequencing (mRNA-Seq) reads from purified caudal epididymal murine sperm cells and peripheral blood leukocytes (Pbl). Fpkm, fragments per kilobase of transcript per million mapped reads. The dotted line shows the expected number of sequence reads for genes with similar expression levels in the sperm cells and Pbl. *CatSper1* and *CatSper2* indicate sperm-specific transcripts; *TRPV2* and *CD37* indicate leukocyte-specific transcripts. Transcripts for *TRPV4, TRPM3,* and *TRPM4* are shown. Raw and processed data can be downloaded from NCBI; with GEO accession number GSE290696. **b** Table shows expression levels of corresponding transcripts. **c** Representative $I_{Cs}^+$ from TRPV4$^{+/+}$ murine sperm (left) show similar potentiation in response to indicated temperatures as $I_{Cs}^+$ from TRPV4$^{-/-}$ murine sperm (right). The baseline corresponds to currents in the HS solution. **d** $I_{Cs}^+$ densities obtained from voltage ramp recordings shown in (**c**) at either 22 °C or 38 °C were independent of the genotype. Thus, TRPV4 is unlikely to affect murine CatSper temperature activation. Specifically, at − 80 mV, current densities (pA/pF; n = 6) were: − 92.05 ± 9.9 (at 24 °C) and -239.95 ± 18.8 (at 38 °C) for TRPV4$^{-/-}$ sperm cells. $I_{Cs}^+$ densities (pA/pF; $n$ = 7) at − 80 mV for wild-type sperm cells were -58.05 ± 5.8 (at 24°C) and -183.07 ± 30.8 (at 38 °C). Similarly, $I_{Cs}^+$ densities at

+ 80 mV (pA/pF; $n$ = 6) for TRPV4$^{-/-}$ cells were 147.84 ± 7.3 (at 24 °C) and 254.13 ± 14.1 (at 38 °C). For wild-type cells at + 80 mV, $I_{Cs}^+$ densities (pA/pF; $n$ = 7) were 175.27 ± 12.4 (at 24 °C) and 298.77 ± 17.8 (at 38 °C). Data are mean values ± S.E.M. Insert: Mouse genotyping is shown. **e** Representative non-CatSper conductance obtained in response to voltage ramps in $I_{CatSper}$ blocking Mg$^{2+}$-containing medium in the presence of either pregnenolone sulfate (PS; TRPM3 agonist, red) or primidone (TRPM3 antagonist) plus PS (green). No change in response to the exposure of either PS or PS + primidone was observed. **f** Representative non-CatSper conductance obtained in response to voltage ramps in $I_{CatSper}$ blocking Mg$^{2+}$-containing medium in the presence of capsaicin (TRPV1 agonist, blue). No appearance of the additional conductance was observed. **g** Non-CatSper monovalent (sodium) current densities (pA/pF) obtained as described in (**e**) and (**f**) and sampled at −80 mV and +80 mV. Specifically, at − 80 mV, current densities (pA/pF) were as follows: − 2.1 ± 0.2 (HS; $n$ = 9); -2.9 ± 0.7 (control, $n$ = 9); − 1.9 ± 0.1 (PS; $n$ = 6); -2.0 ± 0.3 (PS + primidone, $n$ = 6), and -3.3 ± 1.1 (capsaicin, $n$ = 3). At + 80 mV, current densities (pA/pF) were as follows: 4.8 ± 0.4 (HS; $n$ = 9); 13.0 ± 2.3 (control, $n$ = 9); 9.4 ± 0.6 (PS; $n$ = 6); 8.3 ± 0.8 (PS + primidone, $n$ = 6), and 10.9 ± 1.8 (capsaicin, $n$ = 3). Data are mean values ± S.E.M.

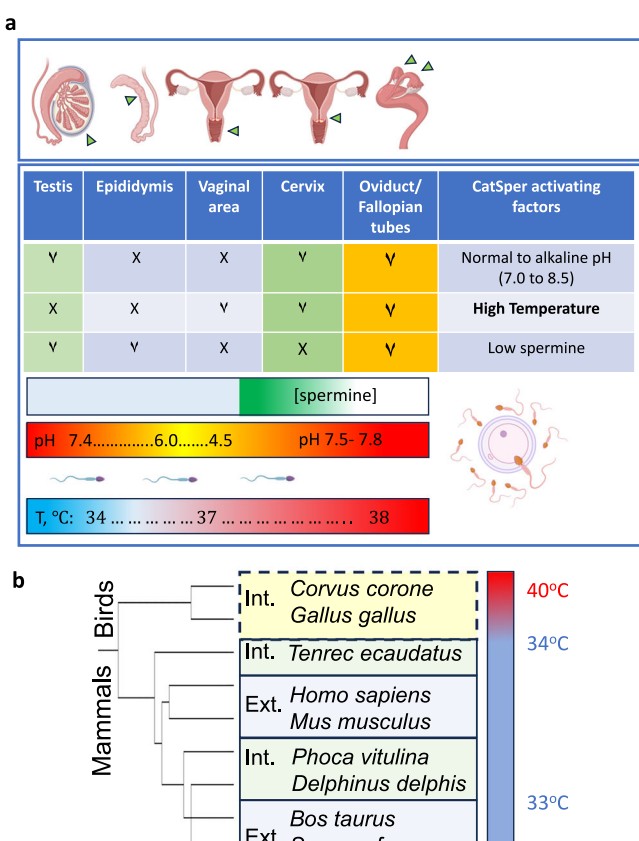

**Fig. 7 | Sperm exposure and adaptation to variable environments. a** Cartoon demonstrates spermatozoon's journey through male and female reproductive tracts and the diverse environments it encounters, including pH, spermine, and temperature gradients. Created in BioRender. Lishko, P. (2025) https://BioRender.com/z91g452. **b** Phylogenetic tree shows species in which CatSper is present (mammals; solid boxes) and in which CatSper is absent (birds; dotted box). Averaged testicular temperatures are indicated, as well as internal (Int) vs external (Ext) testes. *Corvus corone* – Carrion crow; *Gallus gallus*- Red junglefowl (chicken); *Tenrec ecaudatus*- Tailless tenrec; Homo sapiens- human; *Mus musculus*- house mouse; *Phoca vitulina* – Harbor seal; *Delphinus delphis* – short-beaked common dolphin; *Bos taurus* – common cow; *Sus scrofa*- wild boar.

ensure their optimal survival and preserve fertilization competence among mammals.

## Discussion

In most mammals, sperm cells are produced, matured, and stored at a temperature cooler by 2–4 °C in comparison to the core body temperature. To ensure this specialized environment, some mammals developed externalized testes, while certain marsupials with internal testes evolved to lower their core body temperature[43–45]. In many mammals, including humans, testes are acutely sensitive to a temperature increase by only a few degrees, with several studies focusing on understanding the underlying mechanism[46]. It is known that a testicular temperature of 34 °C or below is required to ensure a proper environment for the enzymatic activity of spermatogenesis, while temperatures > 36 °C were shown to be detrimental to sperm fertility[47]. The medical condition varicocele causes an elevation of testicular temperature and impairs sperm motility leading to male infertility[48]. However, the molecular mechanisms of temperature impact on sperm physiology are insufficiently explored. Indeed, spermatozoa eventually complete their journey in the warmer environment of the oviduct without any obvious detrimental effect on their fertility which was previously unexplained.

Spermatozoa are equipped with an array of sensory molecules, including ligand-gated, pH-sensitive, and voltage-dependent ion channels, with CatSper being the primary calcium channel vital for fertilization. By exploring the mechanism of temperature effect on sperm physiology, we found that CatSper is a temperature-gated ion channel with a thermal threshold of 33.5 °C in mice. CatSper activation and consequent calcium influx into sperm flagellum have been shown to activate a self-destruction mechanism via ubiquitination that limits sperm cell survival[15,49], so they have a precise deadline to find and fertilize an egg. Hence, premature CatSper activation, especially its activation in the testes, would be disadvantageous to sperm fertility. Thus, CatSper temperature sensitivity presents a spermatozoan Achilles' heel to render them dysfunctional upon testicular heating.

To prevent this from happening, the male reproductive system built-in a few self-guarding mechanisms (Fig. 7a). The first one is an acidic epididymal environment. After sperm complete their genesis in the testes, they undergo a quiescent storage and maturation phase in the epididymis, where the cauda epididymis plays a major role in the regulation of sperm remodeling to render them capable of fertilization[46,50]. Interestingly, the caput epididymis maintains an acidic pH of ~ 6.5, while cauda epididymis has a pH of 6.7 to 6.8[51]. In addition,

the entire epididymis remains in a cooler environment of the scrotal sac compared to the abdominal counterparts. From our findings, it is now evident that these factors are essential to keep CatSper protected from accidental overheating and to prevent CatSper premature activation in the male body (Fig. 7a). However, after exposure to seminal plasma (pH 7.4) and their consequent deposit into the female reproductive tract that provides a 37–39 °C environment[52], the acidic protection disappears. However, seminal plasma provides the next level of protection to CatSper via one of its major components, spermine. As shown in this study, spermine imposes a strong, fast, and prolonged inhibition on CatSper, by changing its voltage-dependent and temperature-gating properties. Thus, spermine protects CatSper from elevated temperatures. Spermine binding can only be removed during capacitation upon exposure to heparin, which is produced by oviductal epithelia[32,35]. By traveling from the cooler environment of the epididymis to the relatively warmer environment of the female reproductive tract (Fig. 7a), and eventually to even warmer regions of the oviduct, CatSper is primed to exert its maximal potency only when sperm cells are in close vicinity of the egg (Fig. 7a). As sperm cells capacitate, their CatSper temperature sensitivity also decreases. Therefore, spermatozoa's ability to adapt to higher temperatures during capacitation seems to provide prolonged protection and extend their life in the oviduct. It is highly suggestive that such sperm adaptation may be related to CatSper presence.

Hypothetically, evolutionary pressure for testicular internalization with CatSper preservation has offered few options: to lower the core body temperatures as in Madagascar marsupial tenrec[44,53], to develop anatomical features to cool internal testes as in marine mammals[54–56] or to get rid of CatSper altogether as achieved by the avian species[57] (Fig. 7b). In marine mammals, such as seals and whales, testicular internalization was an essential adaptation to protect testes from a cold aquatic environment and ensure better hydrodynamics and predatorial evasions[54]. Testicular internalization in birds may have been followed by similar evolutionary selection, with improved aerodynamics being a strong selection factor (Fig. 7b). Interestingly, many avian species that have elevated core temperatures of 41 °C also have internal testes and, as a result, have testicular temperatures reaching 40 °C without any detrimental effect on sperm motility. That could be explained by the absence of CatSper genes in the avian genome[57].

Overall, CatSper polymodality and adaptability represent a unique molecular evolutionary adaptation to safeguard male germ cells on their long journey through the male and, eventually, female genital tract. Further studies to explore the effect of CatSper inhibiting factors on their ability to de-capacitate sperm could provide an opportunity to develop better sperm storage methods for fertility preservation. Additionally, future studies on CatSper modalities across animal kingdoms, including humans, may provide useful new tools to develop CatSper-based contraceptives and improve human fertility worldwide.

## Methods

Our research complies with all relevant ethical regulations, and all animal research was approved by the WashU Medicine Animal Care and Use Committee (protocol 22-0251).

### Animals

C57BL/6NJ male mice, 3–6 months old, were purchased from Charles River, and C57BL/6 J mice were purchased from Jackson Laboratory (JAX) and housed at the barrier animal facility of Washington University in St. Louis, School of Medicine (WUSM), in a room with a controlled dark-light cycle (10 h dark followed by 14 h of light) and a controlled temperature of approximately 23 ± 0.5 °C. The animals were fed a standard chow diet and provided with chlorinated water *ad libitum*. Euthanasia was performed in accordance with NIH Guidelines

for Animal Research and approved by the WashU Medicine Animal Care and Use Committee (protocol 22-0251), with every effort made to minimize animal suffering. Specifically, euthanasia was carried out using $CO_2$ followed by cervical dislocation. There was no difference in the sperm quality between both lines. In addition to wild-type C57BL/6 N male mice, CatSper1[-/-] (from Dr. Jean-Ju Chung, Yale University) and TRPV4[-/-] (from Dr. Hongzhen Hu, WUSM) male mice were used in this study. TRPV4[-/-] mice were fertile and were bred as TRPV4[-/-]. However, since CatSper1[-/-] mice are known for their male infertility, CatSper1[-/-] males were born at a Mendelian ratio by breeding CatSper1[-/-] females with CatSper1[+/-] males. Genotypes were continuously monitored by TransnetYX, Inc. (TN) and verified by in-house genotyping, as discussed below. For sperm and blood transcriptomes, C57BL/6NHsd male mice were used (Envigo, IN). Overall, 102 male mice were used in this study.

### Genotyping

To minimize animal suffering, ear punches were used to obtain genomic DNA. The genotyping was performed using EmeraldAmp GT PCR Master Mix (Takara Bio USA, RR310) and primers flanking the deleted region of CatSper1 exon 2: Fw: 5-GGAGGGTACCTGGGAT-CACT-3, and Rev: 5-ACACCGGCCTTATTCCAAG-3, were used with Tm = 59 °C and Tm = 53 °C, respectively. Control primers flanking CatSper1 exon 2: Fw: 5-GGAGGGTACCTGGGATCACT-3, and Rev: 5-CAAGGCCCCGTGACTCTTAC-3, were used with Tm = 59 °C. To confirm the genotype of TRPV4[-/-] mice, primers flanking the deleted region of exon 4 of TRPV4: Fw: 5-GCTATTCGGCTATGACTGGG-3, and Rev: 5-CAAGGTGAGATGACAGGAGATC-3, were used with Tm = 57 °C and Tm = 61 °C respectively. Control primers flanking the existing region of exon 4 of TRPV4 were used: Fw: 5- TGTTCGGGGTGGTTTGGCCAGG ATAT-3, and Rev: 5-GGTGAACCAAAGGACACTTGCATAG-3, with Tm = 75 °C and Tm = 69 °C respectively. Each male mouse, positive for the heterozygous deletion of a region of CatSper1, was placed in breeding with CatSper1[-/-] females.

### Chemicals and solutions

Most of all, chemicals were purchased from Sigma-Aldrich and Fisher Scientific. All the reagents used in this study were of the highest purity grade (BioXtra or similar). There were no differences between the actions of these compounds purchased from different vendors. Tris-HCl was obtained from Quality Biologicals, pregnenolone sulfate and NNC 55-0396 from Tocris, Embryo Human Tubal Fluid medium (Embryomax-HTF; MR-070-D) from Millipore. DMSO was used as a solvent for NNC 55-0396 with final DMSO concentrations < 0.1%.

### Murine sperm isolation

The cauda epididymal sperm cells were isolated from euthanized male mice and used for all experiments. All sperm cells were collected immediately after euthanasia and placed in a high saline (HS) solution as described previously[18]. HS solution contained in mM: 135 NaCl, 20 HEPES, 5 KCl, 2 $CaCl_2$, 1 $MgSO_4$, 5 Glucose, 10 Lactic acid, 1 sodium pyruvate, pH 7.4 was adjusted with NaOH; 320 ± 5 mOsm/L.

### Murine sperm capacitation

To obtain capacitated sperm cells, the caudae epididymides were directly collected in an Embryomax-HTF medium and treated as described in ref. 18. After multiple incisions, caudae epididymides were incubated in this capacitation medium initially for 10 min in the $CO_2$ incubator at 37 °C and 5% $CO_2$, after which caudal tissues were discarded, and the remaining released sperm cells were allowed further incubation at 37 °C and 5% $CO_2$ for timed-controlled capacitation between 45 and 90 min. At selected time points, 45 min, 60 min, and 90 minutes, sperm cells were processed for confirmation of capacitation using either electrophysiology or video recording. Capacitated spermatozoa were visually identified based on hyperactivated motility that is demonstrated

as excessive asymmetric flagellar beating. Upon successful break-in and establishing a whole-cell mode, capacitated spermatozoa displayed larger membrane capacitance (2.77 pF to 2.95 pF), in comparison to non-capacitated spermatozoa (2.45 pF to 2.53 pF).

## Electrophysiology

Recordings were performed using the whole-cell patch clamp method as described previously[11,18,19]. Borosilicate glass micropipettes (Sutter Instruments; BF 150-86-7.5) were pulled using a Flaming/Brown micropipette puller (Model P-1000, Sutter Instruments) and fire-polished with Microforge MF-830 (Narishige, Japan). The gigaohm (GΩ) seals were formed between the glass micropipette and sperm cytoplasmic droplet in HS solution. The GΩ seals ranging from 5 to 20 GΩ were selected for break-in, and whole-cell mode was achieved by applying brief mouth suction to pull the sperm membrane into the patch pipette followed by short voltage pulses from 0 mV to 520–670 mV with 1–5 ms duration. After establishing the whole-cell mode of contact, non-leaky sperm cells with seal resistance around 1.5 GΩ, and access resistance of 45–105 mΩ were selected for recording. The membrane capacitance (Cm) ranged from 2.25 to 2.6 pF for non-capacitated and 2.65 to 3.10 pF for capacitated sperm cells. Altogether, 187 non-capacitated and 93 capacitated sperm cells were used for data collection and analysis. The recordings were made using an Axopatch 200B amplifier and an Axon™ Digidata 1550 A digitizer (Molecular Devises, Sunnyvale, CA, USA) integrated with a Humbug noise eliminator, and the hardware was controlled with the Clampex 10.5 software (Molecular Devices, CA). To record monovalent CatSper currents, divalent free medium (DVF) was used as bath solution (in mM): 140 CsMeSO$_3$, 40 HEPES, 1 EDTA, pH was adjusted with CsOH; 320 ± 5 mOsm/L. Cs$^+$-based intracellular pipette solution contained (in mM): 130 Cs- methanesulfonate (CsMeSO$_3$), 70 HEPES, 3 EGTA, 2 EDTA, 1 CsCl; pH was adjusted with CsOH; 330 mOsm/L. The elevated temperatures were applied via a built-in in-line heater (SH-27B, Warner Instrument Corporation, Hamden, CT, USA) equipped with a thermal sensor and connected to a single-channel heater controller (TC-324B, Warner Instrument Corporation, Hamden, CT, USA). The latter was connected to the 1550 A Digidata to ensure simultaneous recordings of the temperature and currents. The connection also ensured a temperature ramp from 22 °C to up to 41 °C. The mini probe detector was positioned in proximity to the recording cell to ensure continuous recording of the exact temperature to which the cell was exposed. For the recording of divalent currents, the following pipette solution was used (in mM): 145 NMDG, 100 HEPES, 10 BAPTA, 0.5 Tris-HCl, pH 7.4 was adjusted using methanesulfonic acid. The bath solution contained in mM: 50 Ba$^{2+}$, 90 NMDG, 20 HEPES, pH 7.4. To subtract baseline, Mg$^{2+}$ based bath solution was used containing in mM: 2 MgCl$_2$, 150 NMDG, 100 HEPES, pH 7.4 adjusted with methane sulphonic acid (HMeSO$_3$). The non-CatSper monovalent current were measured using 140 NaMeSO3, 1 MgCl2, and 20 HEPES, pH 7.4. The signals were acquired at 10 kHz and filtered with a low-pass 1 kHz Bessel filter. Electrophysiology recordings were performed for both non-capacitated and capacitated murine caudal sperm cells. The capacitated sperm cells showing vigorous and asymmetric beating of flagella, and poor attachment with the glass coverslip were selected for patching. The capacitated sperm cells showed the formation of instantaneous GΩ tight seals due to their flexible membrane.

## Data analysis

All the data are presented as mean values ± S.E.M., and the number of individual sperm cells (n) analyzed unless stated otherwise. To calculate current densities, the amplitudes were normalized to capacitance (Cm, pF). Capacitance artifacts were graphically removed during data analysis. Two-way analysis of variance (ANOVA) was done for the determination of the current-temperature relationship, and post-hoc Tukey's test was used for comparison of mean values and statistical significance. Statistical significance is indicated by * $p < 0.05$, ** $p < 0.005$, *** $p < 0.001$, and **** $p < 0.0001$. Data were analyzed using Clampfit 10.7, Origin Pro 9.0, Origin Pro2024b, and Prism 8.0.

## Determination of thermal threshold ($T_h$) and $Q_{10}$

The temperature-activation relationship was determined by plotting the log ($I_{Ba}^{2+} / I_{Ba}^{2+}{}_{min}$) acquired at −20 mV on a log10 scale against the reciprocal of the absolute temperature. After baseline subtraction (baseline acquired in HS), $I_{Ba}^{2+}$ at corresponding temperatures were normalized to the $I_{Ba}^{2+}$ at 22 °C ($I_{Ba}^{2+}{}_{min}$) to obtain the amplitude ratio. Two phases of the thermal response: the initial slow phase and another phase corresponding to the steepest component of the plot, were linear fitted, and the point of their intersection indicated the thermal threshold ($T_h$). The inverse slope of the linear fits indicated corresponding $Q_{10}$. In addition, $Q_{10}$ can be calculated using the following equation: $Q_{10} = [I_2/I_1]^{10 \,°C/(T2−T1)}$, where $I_1$ is $I_{CatSper, Ba}^{2+}$ at the lower temperature ($T_1$), and $I_2$ is the $I_{CatSper, Ba}^{2+}$ at the higher temperature ($T_2$); T is the temperature in degree Celsius.

## RNA-SEQ transcriptome of murine sperm and leukocytes

RNA deep sequencing analysis was performed as described in ref. 58 with modifications. Specifically, total RNA was extracted from murine epididymal spermatozoa isolated from three 6-month-old C57BL/6NHsd male mice. The sperm population was purified by gradient centrifugation using ISolate™ (Irvine Scientific, CA) density gradient. 80% and 40% ISolate™ gradients were prepared with HS solution. Sperm purification was carried out by overlaying epidydimal sperm suspension on top of 40% gradient and centrifuged for 30 min at 300 g. Pure sperm cells were collected at the bottom and visually examined for purity and contamination under a phase-contrast microscope. Only 100% pure sperm samples were used for the RNA extraction. To exclude potential contamination with leukocytes, a comparison of both transcriptomes was used to determine sperm-specific transcripts. Peripheral blood leukocytes (Pbl) were obtained from three 6-month-old C57BL/6NHsd male mice postmortem using heart puncture. The total amount of blood collected was 2 mL. The total RNA was extracted from cells using a Qiagen RNeasy kit and converted to cDNA, followed by fragmentation to 200 bp. The total RNA obtained from murine Pbl was 55.2 μg, and the total RNA obtained from murine sperm was 14.73 μg; both RNA isolations have purity above A260/A280 = 2,1. DNA fragments with appropriate lengths (e.g., 200 bp) were purified using gel electrophoresis, and sequencing libraries were prepared with poly(A)+-enriched RNA using Illumina mRNA-Seq Sample Prep Kit according to the manufacturer's instructions. Illumina Single Read Cluster Generation Kit v2 (Lot 4941271) was used for cluster generation. Final libraries concentrations were: mouse sperm: 49.1 ng/mcL; mouse leukocytes: 41.4 ng/mcL. The final libraries were assessed using a Bioanalyzer to check fragment size, concentration, and purity. The sperm library (well #4) and pbl library (well #5) were loaded onto a 1.4 mm single-read flow cell (42VKHAAXX; P/N 1004224), where they underwent bridge amplification to form clonal clusters. Libraries were sequenced on the Illumina Genome Analyzer II using a 36-cycle Run v3 sequencing kit (Lot 4935209) by standard protocols. Sequences were aligned to the mouse genome GRCm39. Bioinformatics support was provided by Centrillion Biosciences Inc. (Palo Alto, CA). Data can be downloaded from NCBI; GEO accession number is GSE290696.

## TRPM3 expression in HEK293 cells and electrophysiology

For recombinant expression, cells of the human embryonic kidney cell line (HEK293, ATCC, CRL-1573) were plated in a 6-well plate and transfected with 2 ug of pIRES2-eGFP-hTRPM3 construct using Lipofectamine 2000 reagent (Life Technologies, 11668027) according to the manufacturer's recommendation. The enhanced green fluorescent protein of the pIRES2-EGFP vector was also used to visualize the

transfected cells for whole-cell patch clamp experiments 24 h after transfection. Gigaohm seals were established in Krebs Ringer solution, composed of 135 mM NaCl, 5 mM KCl, 1 Mm CaCl2, 1 mM MgSO4, 0.4 Mm KH2PO4, 20 mM HEPES, and 5.5 mM Glucose (310 mOsm, pH 7.3, adjusted with KOH). The pipette buffer was composed of 130 mM CsMeSO3, 20 mM HEPES, 5 mM BAPTA, and 1 mM MgCl2 (295 mOsm, pH 7.3, adjusted with CsOH). The bath solution contained 130 mM NaMeSO3, 43 mM HEPES, and 1 mM MgCl2 (310 mOsm, pH 7.3, adjusted with NaOH). The TRPM3 agonist Pregnalone sulfate (50 μM; Tocris, 5376), alone or in combination with the TRPM3 antagonist Primidone (5 μM; Sigma P7295), was applied directly to the bath solution.

### Video recording of murine sperm motility

Spermatozoa were isolated from caudae epididymides from wild-type male C57BL/6NJ mice 3 months old, as described above. Both non-capacitated and capacitated sperm cells were plated onto 5 mm coverslips in HS solution. Cell motility was recorded immediately after isolation or after timed capacitation using a high-speed GX-1 Memrecam camera (NAC Image Technology) linked to an Olympus DIC IX71 microscope. The speed of recording was 500 frames per second (fps), and video recordings were slowed down to playback at 100 fps.

### Reporting summary

Further information on research design is available in the Nature Portfolio Reporting Summary linked to this article.

## Data availability

All data are available in the manuscript or in the supplementary materials. Raw and processed data are available from NCBI GEO under accession number GSE290696. The data and materials used in the analysis will be available to any researcher for purposes of reproducing or extending the analysis. Source data are provided in this paper.

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

## Acknowledgements

*Funding:* This work was supported by the BJC Investigator fund to P.V.L. We thank Dr. Elena Gracheva for the help with sperm and leukocyte transcriptome preparations and Dr. Boheng Liu for the initial assessment of TRPV4 function in sperm. We also thank Dr. Simon Vu for his advice on ion channel temperature regulations.

## Author contributions

D.K.S. and P.V.L. contributed to the conception and design of this research, data acquisition, and processing and wrote the manuscript. D.K.S. obtained all electrophysiological data. C.V. performed mouse genotyping and biochemistry data acquisition. J.C.A. performed recordings from TRPM3 in HEK293 cells. P.V.L. provided day-to-day supervision of the project, helped with electrophysiological data acquisition, helped with data analysis, and supported the project since its conception. All authors performed/contributed to experiments, analyzed data, contributed to writing, reviewed, and approved the final version of the manuscript.

## Competing interests

The authors have no competing interests.
