## [Peer Review file · Nature Communications]

The essential calcium channel of sperm CatSper is temperature-gated

Corresponding Author: Professor Polina Lishko

Version 0:

Reviewer comments:

Reviewer #1

(Remarks to the Author)

In this manuscript, “The essential calcium channel of sperm CatSper is temperature-gated,” the authors investigate how CatSper ion channels in murine sperm are modulated by temperature, spermine, and pH. Their findings increase understanding of the mechanisms of sperm capacitation in the female reproductive tract, and the mechanisms by which premature capacitation is prevented. Here the authors propose that expression of CatSper channels is required for temperature-evoked currents, that these temperature-evoked currents are partially inhibited by spermine, voltage-shifted after capacitation, and persist in TRPV4-null sperm. We offer critical and suggested changes to improve the accuracy and readability of the authors’ conclusions.

Critical improvements:

- More context should be given so that non-expert readers can easily recognize what is novel about these findings, and how they situate with what is known in the reproductive physiology field. For instance, is this the first time CatSper channels have been shown to be modulated by temperature? How do these findings fit with previous evidence that the TRPV4 channel is an essential temperature sensor in human sperm? Are there species-specific differences, or do the current findings shift the paradigm of temperature sensation in sperm?
- It is inaccurate to refer to the graph in Fig. 1i as an Arrhenius plot. A true Q10 depends on reaction rates and cannot be measured from steady-state currents. If the authors desire to present an Arrhenius plot, they could measure current activation and deactivation rates from current families as in Fig. 1b and 1f. Another possibility is to leave the graph as is—if so, it should not be referred to as an Arrhenius plot and references to Q10 should be removed. Raw data should also be included. The identification of a putative thermal threshold of CatSper has no real biophysical meaning and overinterprets the data.
- The assertions about pH-dependence of CatSper inactivation (lines 105-120) don’t fully adhere to the data. Comparing Ext. Fig. 2b to Fig. 1b, it appears that inactivation is faster at both 22 and 38 °C in pH 6 compared to pH 7.4 rather than “heat exposure resulted in brief CatSper activation followed by a fast inactivation.” In Ext. Fig. 2g, the timescales of temperature and current don’t match. Current responses to discrete temperature steps rather than a ramp would make it easier to compare currents between pH conditions. If a $\Delta V_{1/2}$ is invoked for Ext. Fig. 2e, a statistical comparison between $V_{1/2}$ at 22 and 38 °C should be included—from the overlap of error bars, the difference appears not to be significant.
- The data in Fig. 3f don’t match the summary shown in 3g; the data in 3f look very similar to non-capacitated sperm in Fig. 1f. While the ensemble of channel activity after capacitation is distinct from that of naïve sperm, it’s not obvious that CatSper channels underlie all the changes. To test that hypothesis, pre-capacitance experiments as in Fig. 3 should be performed on CatSper null sperm.

Minor improvements:

- In general, figures will be more navigable if corresponding plots are arranged in parallel rows or columns. For instance, in Fig. 3 panels a-d could form the top row with panels e-h situated directly underneath.
- Consider including a positive control for the CatSper null current recordings in Fig. 1a. Is there another current that could be evoked in these sperm to convince readers that they aren’t dead?
- The data points in Fig. 1d should be made uniform in color and size.
- Statistical comparisons should be added to Fig. 2e and 2f.
- The differences between TRPV4-positive and TRPV4-null sperm look significant in Fig. 4a, yet no statistical difference is observed in 4d. If the summary is based on steady-state data, those data should be shown.
- Consider including a positive control for pregnenolone S and primidone in Fig. 4g.
- Why are TRPV1 and TRPM4 untested as candidate temperature-sensitive channels in Fig. 4?

Reviewer #2

(Remarks to the Author)

This is a timely and extensive investigation about how different aspects of CatSper inhibition are important for sperm function and reproductive success. The authors report that CatSper is highly temperature sensitive, that this heat-activation is attenuated by low pH, that CatSper is inhibited by spermine, and that capacitated spermatozoa loses temperature gating. All these different effects allow the sperm to retain its function and not be maximally activated until it reaches the egg. These are all important findings. However, the text and figures can be drastically improved upon.

Major Comments

1. Some non-standard expressions and abbreviations are used in the text (see minor comments for examples). This makes the text hard to read.
2. The writers assume the reader knows a lot about sperm physiology and many aspects are not explained to un-initiate readers (e.g. minor points 7 and 9).
3. Most current graphs list Y axis as I CatSper, but it is really the current in the spermatozoa (which could contain other currents than CatSper currents). So maybe most prudent to just use I as Y axis label throughout?
1. Figures. In another paper (PMID 30213858) you show (figure 4) a small current in catsper^{-/-}. Did you here see any of those currents in CatSper^{-/-}? Or is the scale too big in fig 1a to see this current? In other words, are the baseline currents 0 in HS solutions, or are the baseline currents the currents after subtracting the HS currents?
4. Not clear where the currents for the I-V curves were measured (e.g. Fig 3c). Please indicate in the figures at what time the currents were measured for the I-V plots.
5. What is Go in your GV plots?
6. The blue dashed boxes are not always explained (e.g. 2d) and not always consistent. For example, why different voltage ranges of blue boxes in 2d and 2f? Fig 3h has a range of -90 to -75mV, whereas the text says -70 mV.
7. Line 135. "spermine exerted a dual action: it inhibited inward ICatSper and shifted midpoint activation upon heat exposure from -85.5 mV (spermine absence) to -25.3 mV (spermine presence)". Are these two actions not the same thing? Shifting the voltage dependence to more positive voltages will remove the inward currents at negative voltages.
2. Fig. 2c. "Spermine exposure did not change IVR at negative membrane potentials". This is not correct: Spermine inhibited currents at negative voltages. Maybe you mean that temperature did not change the currents in the presence of spermine since there is no current at 22 nor 38 degrees?
3. Fig. 2e. The dotted line in 2e is explained in legend to 2f.
4. Fig 3c. Is this statistically different than Fig 1d? And is Suppl Fig 2a different than these?
5. Fig 3i. "No midpoint activation was observed". Probably you mean "no shift in midpoint of activation was observed"?
6. Explain HS and DVF in the first legend when they appear, not just in Methods.
7. The order of panels in the figures does not follow any specific rule (e.g. from left to right a,b,c,d, e/f, g, h, i or a/b. c/d. e/f, g/h). It is not easy for the reader to follow.
8. Fig 2a is the same as Fig1a. Should be removed.
9. Fig 2e and f are not explained in the text
10. Fig 4 d, e, and f are not explained in the text.
11. Fig 4g. NMF does not appear in any other part of the text.
12. Sections are not clearly defined. Text should be divided into sections (Results, Discussion etc) and subsections for clarity
13. No statistics are provided for any data or figure (only in extended figures)
14. I think they overstate the conclusions in the last part of the paper, such as the evolutionary pressure.

Minor Comments

1. Abstract should not include references
2. Introduction is very general. It should include more features about the CatSper channel itself
3. Line 77. IVR is not a standard abbreviation in the field. Usually mentioned as I-V curves (or I-V relationships). IVR usually means Integrated voltage regulators.
4. Line 109. Not clear why low extracellular pH renders intracellular pH low also?
5. Line 114. "No significant increase in ICatSper was observed when measured at steady-state". Maybe insert the words "upon heat exposure" in this sentence after "observed" for clarity.
6. Line 111 Fig. S2a-d, f. Here you use S2 and other places you use Extended Data Fig to describe Supplementary Figure. Please be consistent.
7. Line 130. "Extracellular application of 3 mM spermine within seconds resulted in a complete loss of CatSper temperature sensitivity". Would be clearer as "Within seconds, extracellular application of 3 mM spermine resulted in a complete loss of CatSper temperature sensitivity".
8. Line 132. "that was impossible to reverse even after prolonged washout". Better to state "did not wash out after 8 minutes of wash", which is what you show in Suppl. Fig 3.
9. Line 137. "Additionally, spermine exposure under heat and not at 24 °C profoundly increased ICatSper voltage-dependency by decreasing its slope (k), from 43 to 19." The slope of the G-V is increased but the slope factor k is decreased. In addition, usually one uses voltage dependence, not dependency.
10. Line 150. How was capacitation obtained? No mention of methods and no references. What capacitation means and how it is obtained would be good to state here for uninitiated readers.
11. Line 158. "Midpoint activation for monovalent ions in capacitated spermatozoa retained a midpoint drop from +2.5 mV to -47 mV". Retained compared to what? Usually, a shift in midpoint of activation is referred to as a shift in midpoint, not a drop. No statistics are provided for this statement.

12. Line 185-187. This is a run-on sentence. Please divide up the different parts into separate sentences. "Eventually" seems like the wrong choice to start this sentence (do you need it?). Also, the use of the word "activity" seems misplaced. Maybe use "current" instead, which is what you did not observe?
13. Line 166. Why include the word mature here? It might confuse the readers with the capacitated spermatozoa you talk about in the sentence before. Or is there a reason to include mature at this point? Maybe best to remove mature for clarity?
14. Line 173. When would CatSper be exposed to follicular fluid?
15. From line 223 to 235 references are not shown
16. Suppl. Fig 1b. Remove temp scale above 40 degrees. It is confusing.
17. Suppl. Fig 5c. Are the data NS?
18. Suppl. Fig 6a. This is not a very clear figure. Please make a better one. For example, what is the black triangle in the Sperm box? What is the green color in Testis? Is there a reason for the circle (temp), triangle (pH) and box (sperm) for the three variables? Why not let all three parameters being three horizontal bars with the color showing the change in value?

Reviewer #3

(Remarks to the Author)

Reviewer #4

(Remarks to the Author)

Version 1:

Reviewer comments:

Reviewer #1

(Remarks to the Author)

In their revised manuscript, "The essential calcium channel of sperm CatSper is temperature-gated," the authors have made several important changes that clarify and contextualize their findings. Based on these changes, we suggest one critical and several minor edits.

Critical improvement:

In this revision, the authors present new data from human spermatozoa (Fig. S9). Unfortunately, these data raise more questions than they answer and don't substantiate the authors' conclusions about CatSper in human sperm. The authors assert in the abstract and results sections that CatSper functions as a temperature-gated ion channel in humans as well as mice. To see temperature dependence, one should compare currents at 22°C vs 38°C under the same conditions (progesterone, heparin, etc.). Figure S9 shows that ICs⁺ was larger in untreated spermatozoa at 38°C vs 22°C (blue traces in S9a vs S9b). Therefore, these currents (likely composed of currents from multiple ion channel types) were temperature sensitive. However, progesterone (a putative activator of CatSper) appears to have blocked temperature sensitivity (brown trace in S9a vs pink trace in S9b). Adding heparin (which removes the sperm block to increase current from CatSper channels) blocked the progesterone-elicited currents at 22°C (brown traces in S9a vs S9c) and partially rescued temperature sensitivity in the presence of progesterone (brown trace in S9c vs pink trace in S9d). Under no conditions are the currents measured at 38°C larger than those at 22°C in the presence of progesterone. This suggests either that CatSper is less active at 38°C compared to 22°C, that progesterone doesn't activate CatSper at 38°C, that progesterone has other off-target effects, or that the temperature sensitivity of human sperm doesn't depend on CatSper. These data therefore undermine the authors' conclusions. Without a definitive experiment like showing that blocking CatSper decreases/eliminates the temperature dependence of Cs⁺ currents, we recommend that these data (Fig. S9) be removed from the manuscript and that assertions about human sperm be removed from lines 20, 55, and 275-300. Experiments on human sperm could comprise a follow-up study.

Minor improvements:

- It is conventional for temperatures to be written without a space between the digit and the unit (°C).
- We concur with Reviewer 2 (Major Comment #8) that the effects of sperm are all expressions of the same underlying phenomenon. Selectively inhibiting inward Cs⁺ currents (equivalent to outward rectification) will necessarily shift the voltage dependence and increase the steepness of the slope factor. Lines 200-207 should be adjusted accordingly.
- Figure S2a makes more sense as part of Figure S1; Figure S2b is redundant and can be omitted.

Reviewer #2

(Remarks to the Author)

1/1

The essential calcium channel of sperm CatSper is temperature-gated.

The main text has notably improved. The text is better written, more comprehensive, and clearer compared to the previous version of the manuscript. The authors have followed the reviewers' recommendations and have clearly addressed all the requested questions and suggestions.

Figures have notably improved.

New conducted experiments, like in human spermatozoa, enhance the quality of the results and provide a more comprehensive understanding of CatSper channels and their regulation by temperature.

Reviewer #3

(Remarks to the Author)

Reviewer #4

(Remarks to the Author)

Dear *Nature Communications* Editors,

We would like to thank all Reviewers and Editors for their positive, constructive, and helpful feedback, as well as for their time and efforts. We have significantly revised the manuscript to make it more accessible to a non-expert audience, rearranged the figures, corrected and updated the references, clarified confusing sections, and, importantly, added novel results. Specifically, we have included human sperm recordings and pharmacological validation, which further strengthen our conclusions. Below is our point-by-point response (in italics) to the reviewers' suggestions. Please note that the reference numbers have changed, and the figure order has been updated to reflect the requested changes. The edited parts of the manuscript are highlighted in purple.

REVIEWER COMMENTS

Reviewer #1 (Remarks to the Author):

Overall: "...the authors investigate how CatSper ion channels in murine sperm are modulated by temperature, spermine, and pH. Their findings increase understanding of the mechanisms of sperm capacitation in the female reproductive tract, and the mechanisms by which premature capacitation is prevented. Here the authors propose that expression of CatSper channels is required for temperature-evoked currents, that these temperature-evoked currents are partially inhibited by spermine, voltage-shifted after capacitation, and persist in TRPV4-null sperm. We offer critical and suggested changes to improve the accuracy and readability of the authors' conclusions.

Critical improvements:

- More context should be given so that non-expert readers can easily recognize what is novel about these findings, and how they situate with what is known in the reproductive physiology field. For instance, is this the first time CatSper channels have been shown to be modulated by temperature? How do these findings fit with previous evidence that the TRPV4 channel is an essential temperature sensor in human sperm? Are there species-specific differences, or do the current findings shift the paradigm of temperature sensation in sperm?

A: Thank you for this valuable assessment. We have rewritten the introduction to make it more accessible to non-experts and emphasized the significance of our findings, which indeed shift the paradigm of temperature sensation in sperm cells carrying the CatSper channel. We have added additional data recordings from human sperm to demonstrate that CatSper's temperature sensitivity is a consistent trend.

While TRPV4 in human sperm may contribute to temperature sensitivity and possibly supports sodium or calcium conductance, its density in human sperm is significantly lower than that of CatSper. In murine sperm, however, it is clear now that TRPV4 is dispensable.

It is well known that human CatSper requires progesterone for full activation, but this effect has previously been studied only at room temperature—a condition rarely encountered by sperm

physiologically. Unexpectedly, our data revealed that human CatSper is much less sensitive to progesterone at 38°C. Notably, ejaculated human sperm, due to their method of isolation, are naturally loaded with spermine, which minimizes their heat response. Only after removing spermine from human sperm we were able to observe progesterone activation of CatSper at 38°C and the full effect of heat on human CatSper (Fig. S9d). This finding indicates that spermine removal is essential for progesterone activation of human sperm under physiologically relevant conditions.

- It is inaccurate to refer to the graph in Fig. 1i as an Arrhenius plot. A true Q_{10} depends on reaction rates and cannot be measured from steady-state currents. If the authors desire to present an Arrhenius plot, they could measure current activation and deactivation rates from current families as in Fig. 1b and 1f. Another possibility is to leave the graph as is—if so, it should not be referred to as an Arrhenius plot and references to Q_{10} should be removed. Raw data should also be included. The identification of a putative thermal threshold of CatSper has no real biophysical meaning and overinterprets the data.

A: Thank you for this suggestion. While it is true that 10-degree temperature coefficient Q_{10} depends on reaction rates, it is also influenced by the speed of the temperature ramp and the speed of the cell perfusion, with faster perfusion and ramp resulting in higher Q_{10} numbers (PMID: 12077604; Xu et al., Nature 2002). In our experiments, we tested various perfusion systems and in-line heaters but achieved consistent results only with a relatively slow in-line heating (temperature change rate in 0.3 °C/s). This limitation arose because patched sperm were too fragile to withstand more rapid temperature changes.

As a result, we re-calculated the Q_{10} for CatSper Ba^{2+} currents elicited by voltage ramps (Fig. 2b) using fixed time points corresponding approximately to -20 mV (Fig.2b grey triangle). We did not use steady-state currents shown on Figure 2a for this purpose. An example of this recording is now shown in Figure 2b, with cumulative data/raw data in Figure S2b. We have explained in detail how this determination was performed, and strictly followed a well-published method routinely used in TRP ion channel field to determine Q_{10} , thermal threshold and to build an Arrhenius plot. This procedure has been used to determine such parameters for TRPV1, TRPV3 and other temperature-gated ion channels (PMID: 12077604; Xu et al., Nature 2002; PMID: 21814281 Gracheva et al., Nature 2011).

Specifically, Q_{10} was measured using $I_{Ba^{2+}}$ elicited by the voltage ramps (Fig 2b). The current amplitudes were sampled at a fixed timepoint of 400 ms that roughly corresponds to -20 mV (Fig 2b; grey triangle and Fig. 2f). For CatSper, it can be determined by calculating the inverse slope of a log (normalized $I_{Ba^{2+}}$) plotted against $1/T$, where T is the absolute temperature in K^{-1} . Q_{10} of the channels that are not temperature-gated is typically around 1.5. However, murine sperm subjected to a gradual heating protocols initially displayed slow activation phase with a Q_{10} of 2.7 ± 0.1 ($n = 7$ cells), followed by a steeper phase with a Q_{10} at least of 5.01 ± 0.27 ($n = 7$ cells), (Fig. 2f and Fig S2b). Which is a very conservative estimation. The transition point between these phases, referred to as the thermal threshold (T_h), was determined to be 33.5 °C, which notably corresponds to the temperature of the murine scrotum. While the Reviewer correctly notes that an Arrhenius plot can be constructed using current activation and deactivation rates elicited by voltage steps, this method is incompatible with fragile sperm cells.

Repeated voltage steps severely compromise their seals, rendering this approach impractical. In contrast, sperm tolerate repetitive voltage ramps much better, making this the preferred method for data acquisition. Consequently, we have omitted the term “Arrhenius plot” and instead refer to Figure 2f as “the current-temperature relationship”.

In optimizing the data, we recalculated both Q_{10} and T_h , resulting in a Q_{10} of at least 5.01 and a T_h of 33.5 °C. We retained Q_{10} as a parameter because our methodology for its calculation aligns with established practices in the TRP channel field, as evidenced by prior publications (e.g., PMID: 12077604; Xu et al., Nature, 2002; PMID: 21814281, Gracheva et al., Nature, 2011). Similarly, putative thermal threshold determination using this method as is also a standard practice used in this field. Our data strongly indicate that CatSper functions as a temperature-gated ion channel rather than merely being temperature-sensitive. In the field of temperature-gated ion channels, such as TRP channels, this conclusion cannot be robustly made without determining both Q_{10} and T_h using the methods we employed.

- The assertions about pH-dependence of CatSper inactivation (lines 105-120) don't fully adhere to the data. Comparing Ext. Fig. 2b to Fig. 1b, it appears that inactivation is faster at both 22 and 38 °C in pH 6 compared to pH 7.4 rather than “heat exposure resulted in brief CatSper activation followed by a fast inactivation.” In Ext. Fig. 2g, the timescales of temperature and current don't match. Current responses to discrete temperature steps rather than a ramp would make it easier to compare currents between pH conditions. If a $\Delta V_{1/2}$ is invoked for Ext. Fig. 2e, a statistical comparison between $V_{1/2}$ at 22 and 38 °C should be included—from the overlap of error bars, the difference appears not to be significant.

A: Following the suggested data rearrangement, the experiments on the pH dependence of CatSper temperature gating are now presented in Fig. 3 and Fig. S3. We corrected the timescale in Fig. 3e (formerly Ext. Fig. 2g), which was indeed inaccurate—thank you for pointing this out. A side-by-side comparison of current inactivation under both pH conditions is now provided in Fig. S3, as recommended.

Our analysis reveals no significant change in the inactivation rate of the tail currents regardless of the intracellular pH (Fig S3). However, overall current amplitudes were unsurprisingly notably smaller at acidic pH. Additionally, the $V_{1/2}$ values for both pH conditions overlap and are not statistically different, as now detailed in the manuscript. In conclusion, while heat can still enhance CatSper conductance, acidic pH serves as a potent inhibitor that cannot be overridden by elevated temperatures.

- The data in Fig. 3f don't match the summary shown in 3g; the data in 3f look very similar to non-capacitated sperm in Fig. 1f. While the ensemble of channel activity after capacitation is distinct from that of naïve sperm, it's not obvious that CatSper channels underlie all the changes. To test that hypothesis, pre-capacitance experiments as In Fig. 3 should be performed on CatSper null sperm.

A: Data from Figure 3 are now presented as Figure 5. We have included a different representative recording that better aligns with averaged Ba^{2+} conductance of capacitated sperm cells. This is now shown as Fig.5e that represents Ba^{2+} currents in pA. The panel 5f (former Fig 3g) now shows current densities, which is pA/pF, measured at the peak (as indicated by yellow triangle). As has been shown on Fig.1 and S1, CatSper-null sperm produce only few

pA of detectable Cs⁺ current. In this regard, divalent conductance of CatSper-null sperm is negligible (also in Kirichok et al., Nature 2006). Additionally, CatSper1^{-/-} sperm do not undergo normal capacitation (PMID: 29953592;PMCID: PMC6185779;DOI: 10.1002/jcp.26883). Given the fact that such sperm fail to capacitate normally and do not produce any detectable Ba²⁺ currents, and the fact that temperature activation of capacitated sperm is smaller than in noncapacitated sperm, we do not see strong rationale for attempting to capacitate CatSper null sperm.

Minor improvements:

- In general, figures will be more navigable if corresponding plots are arranged in parallel rows or columns. For instance, in Fig. 3 panels a-d could form the top row with panels e-h situated directly underneath.

A: Thank you for these excellent suggestions. To align with the Nature Communications portrait format, we have rearranged the figures, combining the suggested order with the journal's formatting requirements. This adjustment has significantly enhanced the overall readability.

- Consider including a positive control for the CatSper null current recordings in Fig. 1a. Is there another current that could be evoked in these sperm to convince readers that they aren't dead?

A: Figure S1b now includes a zoomed-in view of CatSper1 KO recordings, along with cumulative data highlighting the residual non-CatSper conductance. Figure 1b displays step recordings from CatSper-null sperm, confirming that these cells are viable. They exhibit comparable membrane capacitance (C_m; Fig. S1c), similar break-in characteristics, and the residual non-CatSper conductance typically observed in CatSper-null recordings. Additionally, we include non-CatSper conductance measured with different monovalent solutions, i.e. Na⁺, K⁺, Cs⁺ (Figure S2a), showing that while CatSper is absent from CatSper-null sperm, residual conductances and non-CatSper currents are present, confirming the cell is not dead.

- The data points in Fig. 1d should be made uniform in color and size.

A: All done. Thanks.

- Statistical comparisons should be added to Fig. 2e and 2f.

A: Statistics have been added to Fig S4e (former fig. 2e) and Fig. 4h (former fig. 2f)

- The differences between TRPV4-positive and TRPV4-null sperm look significant in Fig. 4a, yet no statistical difference is observed in 4d. If the summary is based on steady-state data, those data should be shown.

A: The summary is based on current-densities obtained from ramp recordings at -80mV or +80mV at the peaks. We have included different representative cells that resembles WT recordings. Statistically, there was no difference between temperature activation of CatSper in wild-type or TRPV4-null sperm. The initial TRPV4-null recording appears to be smaller, because the control current also is smaller in this cell in comparison to the WT sperm.

- Consider including a positive control for pregnenolone S and primodone in Fig. 4g.

A: Thank you for this excellent suggestion. Positive control has been included as Figure S8. We used human TRPM3 recombinantly expressed in HEK293 cells to show that both pregnenolone sulfate and primidone are active.

- Why are TRPV1 and TRPM4 untested as candidate temperature-sensitive channels in Fig. 4?

A: We tested for the presence of TRPV1 in mouse sperm by applying capsaicin (Figure 6e-g), and as expected, no response was observed. TRPM4, a sodium channel activated by intracellular calcium, was not expected to show significant conductance in our experiments due to the heavy buffering of calcium in our pipette solutions using BAPTA and/or EDTA/EGTA. While it is possible that other non-CatSper temperature-regulated conductance exist in mouse sperm, robust experiments with CatSper-null sperm demonstrate that these channels have only a minor effect. Most of the temperature-activated conductance is predominantly carried by CatSper.

Reviewer #2 (Remarks to the Author):

Overall: This is a timely and extensive investigation about how different aspects of CatSper inhibition are important for sperm function and reproductive success. The authors report that CatSperm is highly temperature sensitive, that this heat-activation is attenuated by low pH, that CatSper is inhibited by spermine, and that capacitated spermatozoa loses temperature gating. All these different effects allow the sperm to retain its function and not be maximally activated until it reaches the egg. These are all important findings. However, the text and figures can be drastically improved upon.

Major Comments

1. Some non-standard expressions and abbreviations are used in the text (see minor comments for examples). This makes the text hard to read.

A: We have removed and clarified those as suggested.

2. The writers assume the reader knows a lot about sperm physiology and many aspects are not explained to un-initiate readers (e.g. minor points 7 and 9).

A: Thank you for this valuable suggestion. As stated above, we have made the text more readable for non-experts.

3. Most current graphs list Y axis as I CatSper, but it is really the current in the spermatozoa (which could contain other currents that CatSper currents). So maybe most prudent to just use I as Y axis label throughout?

A: We have replaced I_{CatSper} with corresponding currents abbreviations, such as I_{Cs^+} , I_{Na^+} or $I_{\text{Ba}^{2+}}$ in all Y-axes for clarity. Although, as evident from experiments with CatSper1 KO mice, 99% of Cs^+ - currents recorded under these conditions are carried by CatSper channels.

4. Figures. In another paper (PMID 30213858) you show (figure 4) a small current in catsper-/- . Did you here see any of those currents in CatSper -/-? Or is the scale too big in fig 1a to see this current? In other words, are the baseline currents 0 in HS solutions, or are the baseline currents the currents after subtracting the HS currents?

A: Indeed, the “baseline” current in former Fig. 1a (now Fig. 1c) is in HS solution. We have now labeled it as “HS” for clarity. Such currents, recorded under good seal conditions usually do not exceed few (1-5) pA at -80mV. Most of the time any larger currents than that would attribute to the component of the “leak” currents. If we understand correctly, the reviewer refers to the paper by Orta et al., JBC 2018 (PMID 30213858), which is not our work. In our hands, the largest non-CatSper conductance that we can observe from CatSper1-/- sperm cells at -80mV does not exceed ~3 pA/pF in HS solution, and is no larger than 10pA/pF in DVF at 38C (Figure S1b). If we compare the scales on the Figure 1e with S1a and S1b, it will be clear that the scale is too big to see this residual component.

5. Not clear where the currents for the I-V curves were measured (e.g. Fig 3c). Please indicate in the figures at what time the currents were measured for the I-V plots.

A: We have indicated by placing colored triangles at which time points the currents were measured.

6. What is Go in your GV plots?

A: Apologies for this typo. We have replaced it with Gmax.

7. The blue dashed boxes are not always explained (e.g. 2d) and not always consistent. For example, why different voltage ranges of blue boxes in 2d and 2f? Fig 3h has a range of -90 to -75mV, whereas the text says -70 mV.

A: we have made it more consistent.

8. Line 135. “spermine exerted a dual action: it inhibited inward ICatSper and shifted midpoint activation upon heat exposure from -85.5 mV (spermine absence) to -25.3 mV (spermine presence)”. Are these two actions not the same thing? Shifting the voltage dependence to more positive voltages will remove the inward currents at negative voltages.

A: It has been clarified as follows: “Spermine exerted a triple action: firstly, it inhibited inward I_{Cs+} (Fig. 4a, 4d). Secondly, spermine shifted midpoint activation upon heat exposure from -85.5 mV (spermine absence; Fig. 1g) to -25.3 mV (spermine presence; Fig. 4f), putting CatSper outside of its activation zone (Fig. 4f, S4f). Thirdly, spermine exposure under heat profoundly increased CatSper voltage-dependence by decreasing its slope factor (k), from 42.6 to 19.2 (Fig. 1g and 4f).”

9. Fig. 2c. “Spermine exposure did not change IVR at negative membrane potentials”. This is not correct: Spermine inhibited currents at negative voltages. Maybe you mean that temperature did not change the currents in the presence of spermine since there is no current at 22 nor 38 degrees?

A: Apologies for this confusion. We have rewritten the entire paragraph to make it clearer. We have included sentence like: “This is also evident from stronger I_{Cs+} outward rectification in the presence of spermine at 38 °C (Fig 4g).”

10. Fig. 2e. The dotted line in 2e is explained in legend to 2f.

A: we removed the dotted lines and clarified legends

11. Fig 3c. Is this statistically different than Fig 1d? And is Suppl Fig 2a different than these?

A: Suppl Fig 2a has been replaced. Statistic has been provided. We have compared now I-V-curves for both pHi 6.0 and 7.4 and included this Figure as S3.

12. Fig 3i. “No midpoint activation was observed”. Probably you mean “no shift in midpoint of activation was observed”?

A: correct. It is now written: “Moreover, midpoint activation shift for divalent ions was even less dramatic: with a tiny change from -9.1 to -9.6 mV (Fig. 5j; S6e).”

13. Explain HS and DVF in the first legend when they appear, not just in Methods.

A: Noted. We have included the explanation at the beginning of the “Result” section.

13. The order of panels in the figures does not follow any specific rule (e.g. from left to right a,b,c,d, e/f, g, h, I or a/b. c/d. e/f, g/h). It is not easy for the reader to follow.

A: The order has been adjusted and made more consistent.

14. Fig 2a is the same as Fig1a. Should be removed.

A: Done.

15. Fig 2e and f are not explained in the text

A: Corrected.

10. Fig 4 d, e, and f are not explained in the text.

A: Corrected.

11. Fig 4g. NMF does not appear in any other part of the text.

A: Replaced with “control” condition and now is Fig. 6g

12. Sections are not clearly defined. Text should be divided into sections (Results, Discussion etc) and subsections for clarity

A: Done.

13. No statistics are provided for any data or figure (only in extended figures)

A: Done. Statistical analyses have been provided

14. I think they overstate the conclusions in the last part of the paper, such as the evolutionary pressure.

A: We have modified discussion to mention evolutionary pressure as a hypothesis, not a proven fact.

Minor Comments

1. Abstract should not include references

A: Indeed. We have removed references.

2. Introduction is very general. It should include more features about the CatSper channel itself

A: Introduction has been rewritten and expanded.

3. Line 77. IVR is not a standard abbreviation in the field. Usually mentioned as I-V curves (or I-V relationships). IVR usually means Integrated voltage regulators.

A: Noted. We have replaced all IVR abbreviations with I-V curves.

4. Line 109. Not clear why low extracellular pH renders intracellular pH low also?

A: we have clarified this in text: "The external acidification usually precedes internal acidification, as sperm adjust their pH to the external environment (Gatti, Chevrier et al. 1993). Therefore, we explored whether the heat-activation of CatSper is preserved under intracellular acidification (Fig. 3)"

5. Line 114. "No significant increase in ICatSper was observed when measured at steady-state". Maybe insert the words "upon heat exposure" in this sentence after "observed" for clarity.

A: Done, thank you for this suggestion!

6. Line 111 Fig. S2a-d, f. Here you use S2 and other places you use Extended Data Fig to describe Supplementary Figure. Please be consistent.

A: Sure thing! According to Nat Comm standards we now use Fig [X], Fig S[X] throughout the manuscript and "extended data terminology" has been omitted.

7. Line 130. "Extracellular application of 3 mM spermine within seconds resulted in a complete loss of CatSper temperature sensitivity". Would be clearer as "Within seconds, extracellular application of 3 mM spermine resulted in a complete loss of CatSper temperature sensitivity".

A: Done, thank you for this suggestion!

8. Line 132. "that was impossible to reverse even after prolonged washout". Better to state "did not wash out after 8 minutes of wash", which is what you show in Suppl. Fig 3.

A: Corrected, thanks!

9. Line 137. "Additionally, spermine exposure under heat and not at 24 °C profoundly increased ICatSper voltage-dependency by decreasing its slope (k), from 43 to 19." The slope of the G-V is increased but the slope factor k is decreased. In addition, usually one uses voltage dependence, not dependency.

A: Corrected, sorry for the typo.

10. Line 150. How was capacitation obtained? No mention of methods and no references. What capacitation means and how it is obtained would be good to state here for uninitiated readers.

A: It is now explained in the text and Methods.

11. Line 158. "Midpoint activation for monovalent ions in capacitated spermatozoa retained a midpoint drop from +2.5 mV to -47 mV". Retained compared to what? Usually, a shift in midpoint of activation is referred to as a shift in midpoint, not a drop. No statistics are provided for this statement.

A: It has been replaced with "Midpoint activation for monovalent ions in capacitated spermatozoa retained a midpoint leftward shift from +2.5 mV to -47 mV upon heat exposure". The data are provided as mean values +/-SEM. They are very different as evident from non-overlapping G-V curves. However, we added statistical analysis for the shaded areas.

12. Line 185-187. This is a run-on sentence. Please divide up the different parts into separate sentences. “Eventually” seems like the wrong choice to start this sentence (do you need it?). Also, the use of the word “activity” seems misplaced. Maybe use “current” instead, which is what you did not observe?

A: Thanks. Has been modified as suggested.

13. Line 166. Why include the word mature here? It might confuse the readers with the capacitated spermatozoa you talk about in the sentence before. Or is there a reason to include mature at this point? Maybe best to remove mature for clarity?

A: Thanks. The word “mature” has been removed.

14. Line 173. When would CatSper be exposed to follicular fluid?

A: Replaced with “upon exposure to the factors provided by the ovulated egg or cumulus cells”

15. From line 223 to 235 references are not shown

A: Corrected.

16. Suppl. Fig 1b. Remove temp scale above 40 degrees. It is confusing.

A: Done. It is now SIe-f.

17. Suppl. Fig 5c. Are the data NS?

A: yes, it is now SIb, NS has been added.

18. Suppl. Fig 6a. This is not a very clear figure. Please make a better one. For example, what is the black triangle in the Spermine box? What is the green color in Testis? Is there a reason for the circle (temp), triangle (pH) and box (spermine) for the three variables? Why not let all three parameters being three horizontal bars with the color showing the change in value?

A: We have provided a better figure, now Figure 7.

Reviewer #3 (Remarks to the Author):

A: Thank you for your constructive and helpful feedback, as well as for your time and efforts.

Reviewer #4 (Remarks to the Author):

A: Thank you for your constructive and helpful feedback, as well as for your time and efforts.

We would like to thank again all Reviewers for their constructive and helpful feedback, as well as for their time and efforts. We have revised the manuscript according to the suggestions. Below is our point-by-point response (in italics). Please note that the figure order has been updated to reflect the requested changes. The major edited parts of the manuscript could be seen with track changes. The large RNA-seq data file has been uploaded to NCBI and accession number has been obtained.

REVIEWER COMMENTS

Reviewer #1 (Remarks to the Author):

Overall: "...In their revised manuscript, "The essential calcium channel of sperm CatSper is temperature-gated," the authors have made several important changes that clarify and contextualize their findings. Based on these changes, we suggest one critical and several minor edits.

Critical improvements:

In this revision, the authors present new data from human spermatozoa (Fig. S9). Unfortunately, these data raise more questions than they answer and don't substantiate the authors' conclusions about CatSper in human sperm. The authors assert in the abstract and results sections that CatSper functions as a temperature-gated ion channel in humans as well as mice. To see temperature dependence, one should compare currents at 22°C vs 38°C under the same conditions (progesterone, heparin, etc.). Figure S9 shows that IC_{s+} was larger in untreated spermatozoa at 38°C vs 22°C (blue traces in S9a vs S9b). Therefore, these currents (likely composed of currents from multiple ion channel types) were temperature sensitive. However, progesterone (a putative activator of CatSper) appears to have blocked temperature sensitivity (brown trace in S9a vs pink trace in S9b). Adding heparin (which removes the spermine block to increase current from CatSper channels) blocked the progesterone-elicited currents at 22°C (brown traces in S9a vs S9c) and partially rescued temperature sensitivity in the presence of progesterone (brown trace in S9c vs pink trace in S9d). Under no conditions are the currents measured at 38°C larger than those at 22°C in the presence of progesterone. This suggests either that CatSper is less active at 38°C compared to 22°C, that progesterone doesn't activate CatSper at 38°C, that progesterone has other off-target effects, or that the temperature sensitivity of human sperm doesn't depend on CatSper. These data therefore undermine the authors' conclusions. Without a definitive experiment like showing that blocking CatSper decreases/eliminates the temperature dependence of $Cs+$ currents, we recommend that these data (Fig. S9) be removed from the manuscript and that assertions about human sperm be removed from lines 20, 55, and 275-300. Experiments on human sperm could comprise a follow-up study.

A: We agree with this assessment. Human data has been removed from the manuscript and will be included in a follow-up study on which we are currently actively working.

Minor improvements:

- It is conventional for temperatures to be written without a space between the digit and the unit (°C).

A: Thank you for pointing this out. We have corrected this throughout the manuscript.

- We concur with Reviewer 2 (Major Comment #8) that the effects of spermine are all expressions of the same underlying phenomenon. Selectively inhibiting inward Cs⁺ currents (equivalent to outward rectification) will necessarily shift the voltage dependence and increase the steepness of the slope factor. Lines 200-207 should be adjusted accordingly.

A: These lines have been re-written as suggested.

- Figure S2a makes more sense as part of Figure S1; Figure S2b is redundant and can be omitted.

A: Figure S2a has been moved to S1 and is now panel S1d. All legends have been corrected accordingly. We have decided to keep S2b (now S2) as it presents individual data distribution as required by the journal rules.

Reviewer #2 (Remarks to the Author):

Overall: The main text has notably improved. The text is better written, more comprehensive, and clearer compared to the previous version of the manuscript. The authors have followed the reviewers' recommendations and have clearly addressed all the requested questions and suggestions. Figures have notably improved. New conducted experiments, like in human spermatozoa, enhance the quality of the results and provide a more comprehensive understanding of CatSper channels and their regulation by temperature.

A: Thank you for your constructive and helpful feedback, as well as for your time and efforts.

Reviewer #3 (Remarks to the Author):

A: We are very grateful for the helpful feedback.

Reviewer #4 (Remarks to the Author):

A: We are very grateful for the helpful feedback.